# 🔗 LINC: A Neurosymbolic Approach for Logical Reasoning by Combining Language Models with First-Order Logic Provers

**Theo X. Olausson*[1]**    **Alex Gu*[1]**    **Benjamin Lipkin*[2]**    **Cedegao E. Zhang*[2]**
**Armando Solar-Lezama[1]**    **Joshua B. Tenenbaum[1,2]**    **Roger Levy[2]**
{theoxo, gua, lipkinb, cedzhang}@mit.edu
[1]MIT CSAIL [2]MIT BCS
*Equal contribution.*

## Abstract

Logical reasoning, i.e., deductively inferring the truth value of a conclusion from a set of premises, is an important task for artificial intelligence with wide potential impacts on science, mathematics, and society. While many prompting-based strategies have been proposed to enable Large Language Models (LLMs) to do such reasoning more effectively, they still appear unsatisfactory, often failing in subtle and unpredictable ways. In this work, we investigate the validity of instead reformulating such tasks as modular neurosymbolic programming, which we call LINC: Logical Inference via Neurosymbolic Computation. In LINC, the LLM acts as a semantic parser, translating premises and conclusions from natural language to expressions in first-order logic. These expressions are then offloaded to an external theorem prover, which symbolically performs deductive inference. Leveraging this approach, we observe significant performance gains on FOLIO and a balanced subset of ProofWriter for three different models in nearly all experimental conditions we evaluate. On ProofWriter, augmenting the comparatively small open-source StarCoder+ (15.5B parameters) with LINC even outperforms GPT-3.5 and GPT-4 with Chain-of-Thought (CoT) prompting by an absolute 38% and 10%, respectively. When used with GPT-4, LINC scores 26% higher than CoT on ProofWriter while performing comparatively on FOLIO. Further analysis reveals that although both methods on average succeed roughly equally often on this dataset, they exhibit distinct and complementary failure modes. We thus provide promising evidence for how logical reasoning over natural language can be tackled through jointly leveraging LLMs alongside symbolic provers. All corresponding code is publicly available.[1]

---

* Author order randomized; all reserve the right to list their name first.
[1] https://github.com/benlipkin/linc

## 1 Introduction

Widespread adoption of large language models (LLMs) such as GPT-3 (Brown et al., 2020), GPT-4 (OpenAI, 2023), and PaLM (Chowdhery et al., 2022) have led to a series of remarkable successes in tasks ranging from text summarization to program synthesis. Some of these successes have encouraged the hypothesis that such models are able to flexibly and systematically reason (Huang and Chang, 2022), especially when using prompting strategies that explicitly encourage verbalizing intermediate reasoning steps before generating the final answer (Nye et al., 2021; Wei et al., 2022; Kojima et al., 2022; Wang et al., 2023b). However, this reasoning ability appears to be unreliable for tasks that require reasoning out of domain (Liang et al., 2022; Saparov et al., 2023), understanding negation (Anil et al., 2022), and following long reasoning chains (Dziri et al., 2023). Furthermore, while the standard approach of "scaling up" seems to improve performance across some reasoning domains, other domains, e.g., reasoning involving use of Modus Tollens, show no such improvements (McKenzie et al., 2022). These findings suggest that such models may be relying on approximate heuristics based on surface-level statistical patterns in reasoning tasks, rather than consistent, generalizable representations and strategies (Srivastava et al., 2023; Creswell et al., 2023).

At the same time, the ability to accurately and soundly perform logical reasoning is important for AI and NLP due to its impact on downstream tasks. For example: retrieval-augmented chatbots may become more truthful if it can be verified that their answers logically follow from the retrieved facts; data-driven models capable of logical reasoning may speed up progress across mathematics and the sciences through automated theorem proving and knowledge discovery; and AI tutoring systems which ensure internal logical consistency might

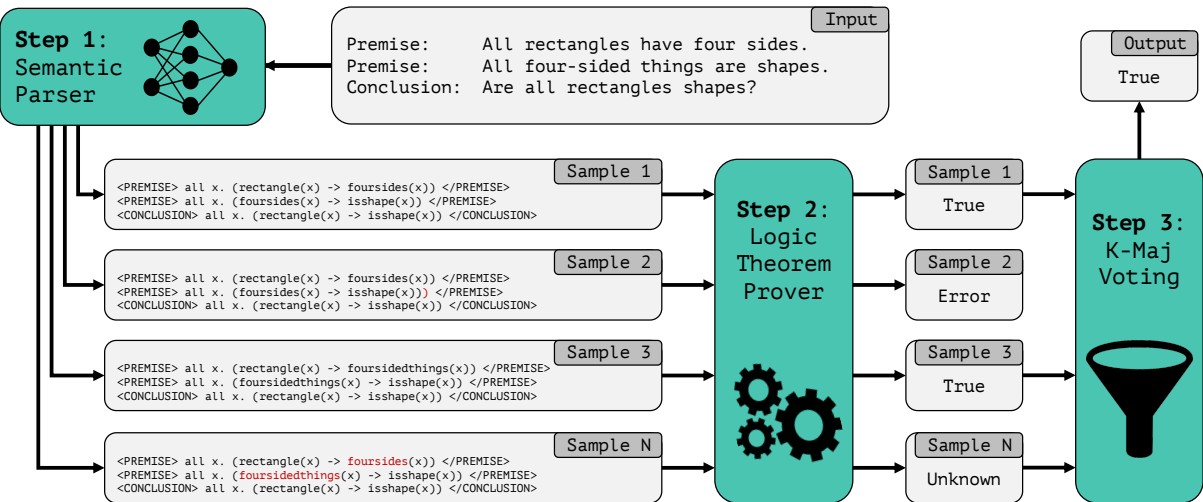

Figure 1: This figure showcases the essence of our approach. Starting from a problem in natural language, in **Step 1**, the LLM semantic parser samples logic formulas expressing estimates of the semantics. It is possible that some of these might contain errors, e.g., the second example shows a syntax error involving an extra parenthesis, whereas the fourth example highlights a semantic error caused by mismatched predicates. In **Step 2**, these are then each offloaded to an automated theorem prover, filtering out syntax errors, and producing labels for the remaining samples. In **Step 3**, the remaining candidate outputs are passed through a majority-vote sieve to arrive at the best estimate for a single output label.

make for better educational platforms, teaching students to think more clearly and rigorously. The question of how to enable state-of-the-art LLMs to become more reliable logical reasoners is thus one of great importance, with far-reaching implications.

In this work, we analyze `LINC`: **L**ogical **I**nference via **N**eurosymbolic **C**omputation (Fig. 1). In `LINC`, logical reasoning is tackled through a modular, two-step neurosymbolic process. First, the language model converts the natural language premises and desired conclusion into first-order logic (FOL) expressions (Enderton, 2001; Barker-Plummer et al., 2011). Second, a symbolic FOL theorem prover algorithmically determines the truth value of the conclusion given the formalized premises. In practice, we also incorporate a third majority voting step, which is shown to improve performance. `LINC` is a natural extension of recent work augmenting lanugage models with symbolic tools such as calculators or interpreters (Schick et al., 2023).

`LINC` has a key advantage: the language model itself no longer needs to perform any deductive reasoning, which is offloaded to the theorem prover. However, there are also clear drawbacks: the formalization from natural language to first-order logic must perfectly capture all relevant information contained in the premises, and any loss of information in the formalization procedure may lead

the solver astray, leading to an incorrect conclusion. As it is not clear whether the task of formalization is more or less difficult than that of end-to-end natural language reasoning, our core interest in this work is to compare and contrast our neurosymbolic approach to existing reasoning strategies like `Chain-of-Thought`. Our contributions are thus three-fold:

- First, we propose `LINC`, a two-stage neurosymbolic approach for logical reasoning tasks (Sec. 2).

- Second, we compare `LINC` to three baseline LLM strategies (Fig. 2), across three models (StarCoder+, GPT-3.5, GPT-4) and two datasets (FOLIO and ProofWriter) (Sec. 4). We find that `LINC` significantly improves performance over every baseline in all experimental conditions except for GPT-4 on FOLIO.

- Third, we provide a thorough error analysis of both `LINC` and `Chain-of-Thought`, identifying three high-level failure modes of each. We discover that these failure modes are distinct, highlighting the potential for a synergy of the two methods (Sec. 5).

Overall, we present strong evidence for the potential of future neurosymbolic logical reasoning systems based on integrating language models and theorem provers.

## 2 `LINC`: Logical Inference via Neurosymbolic Computation

Our neurosymbolic approach to end-to-end logical reasoning consists of two stages. In the first stage, the LLM acts as a semantic parser, translating NL statements into FOL expressions in our supported logic language. In the second stage, these expressions are parsed from the text generated by the LLM and then get passed to an automated theorem prover; we use `Prover9`, a high-performance prover widely used in the logic community (McCune, 2005–2010). The external solver then executes a symbolic deduction algorithm, which either returns a value from the set {`True`, `False`, `Uncertain`} or raises an exception due to improper FOL syntax (e.g., if the model fails to balance parantheses in the formulae).

At its core, the strength of this approach lies in the reformulation of the problem space. End-to-end NL-based reasoning allows for operation over a highly flexible expression space, but leaves the LLM with the difficult task of performing explicit deductive inference over expressions in this space. Using `LINC`, we instead trade off the flexible expression space of NL for syntactically strict logic formulas, allowing us to leverage symbolic algorithms with provable guarantees that the deductive chains will be correct with respect to the semantics of the intermediate representation. Making effective use of this reformulation thus requires the logic expressions generated by the LLM to be 1) *syntactically* valid, such that they are accepted by the prover, and 2) *semantically* valid, such that their evaluation results in the correct conclusion. In our experiments, we mitigate these risks by using a $K$-way majority voting procedure, which is discussed further in Sec. 3.

The significance of this problem space reformulation can—beyond the numerical increases in performance observed across our experiments—perhaps best be seen through an in-depth comparison of how `LINC` and traditional end-to-end LLM reasoning approaches such as `Chain-of-Thought` (`CoT`) *fail*. To foreshadow our latter analysis, we find that compared to `CoT`, `LINC` has worse recall but better precision on `True`/`False` predictions. We discuss this further in Sec. 5 and highlight that this suggests that `LINC`, as well as neurosymbolic computation more generally, has the potential to reduce LLM overconfidence and hallucination.

## 3 Experiments

In this section, we present our experimental setup, the models we use, and the three baselines to which we compare `LINC`.

**Datasets:** Our experiments use tasks from two existing datasets: **FOLIO** (Han et al., 2022) and **ProofWriter** (Tafjord et al., 2021), both of which have been shown to be challenging for off-the-shelf LLMs (Han et al., 2022; Creswell et al., 2023). FOLIO is an expert-written, open-domain, logically complex and diverse dataset for natural language reasoning with first-order logic. We use its validation set for our evaluation. However, of the 204 samples in the validation set, we discover that 22 have errors (details in Appendix C), leaving us with 182 examples for our evaluation. ProofWriter, meanwhile, is a synthetically generated dataset for logical reasoning over natural language. For our evaluation, we use the OWA (Open-World Assumption) portion of ProofWriter, since this setting best matches that of FOLIO. Since we are running a large number of experiments, we randomly select 360 data points to evaluate on in order to reduce costs. We sample these in such a way that the resulting data set is balanced across both the number of reasoning steps in the shortest ground truth proof (depth 0-5; 50 samples each) and across the three labels (`True`/`False`/`Uncertain`; 120 samples each; 20 each per depth).

**In-context learning examples:** We hand-pick eight diverse samples from the FOLIO training set to be used as few-shot in-context examples. Because ProofWriter does not come with ground truth FOL statements, we use these eight samples for both evaluations. Compared to FOLIO, questions in ProofWriter generally have more premises per question (in our validation sets: an average of 5.3 in FOLIO vs. 18.8 in ProofWriter). Thus, our evaluation on FOLIO is an *in-distribution* task, whereas ProofWriter requires generalizing *out-of-distribution* to reasoning over considerably larger sets of premises than are given in the prompt.

**Majority voting:** $K$-way majority voting, in which $K$ samples are taken i.i.d. from the model and the mode is used as the final prediction, has previously been shown to improve the performance of prompting-based strategies in logical reasoning tasks (Wang et al., 2023b). We implement such a strategy in our work, with reported accuracies reflecting $K{=}10$-way majority voting, unless otherwise stated. In the case of ties between two la-

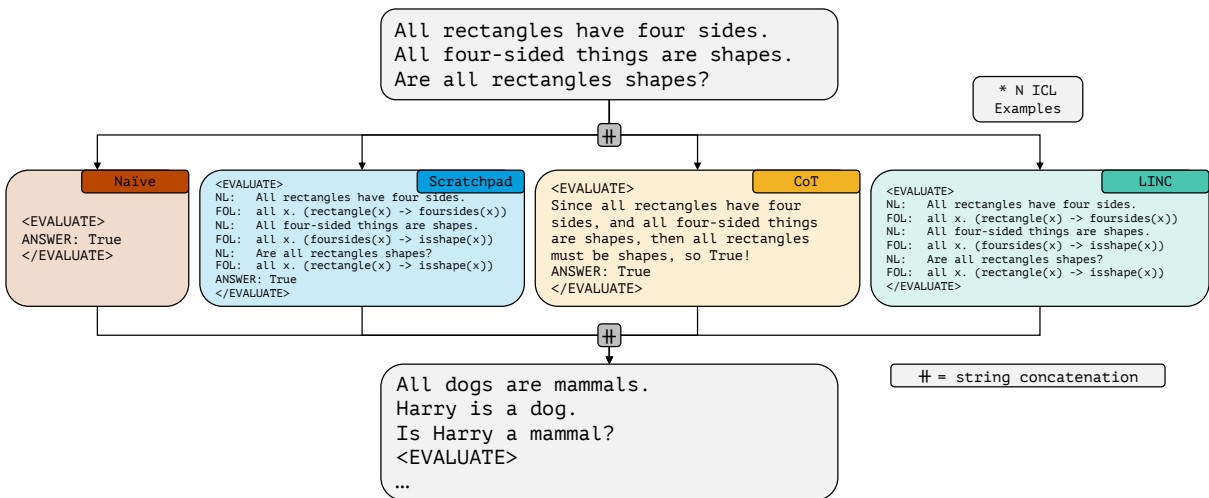

Figure 2: This figure outlines the string concatenation workflow for each of our conditions. We start with the original problem, provide ICL examples through an intermediate markup language, and finally append the problem to evaluate. At this stage, we allow the model to autoregressively sample until producing a stop token.

bels, we arbitrarily select the first of the two to have been generated. We report the effect of $K$ on performance across our conditions and briefly discuss trends in Appendix H.

**Models:** We use three models pre-trained on both natural language and code: GPT-3.5 (Ouyang et al., 2022), GPT-4[2] (OpenAI, 2023), and Star-Coder+[3] (Li et al., 2023) with a decoding temperature of $T = 0.8$ for all experiments. We defer model, hyperparameter, and hardware details to Appendix B. We opt for StarCoder+ for three reasons: firstly, unlike the other models we consider, it is a free, open-access model. Secondly, it has a dataset search functionality[4], with which we verify that FOLIO and ProofWriter are not in StarCoder+'s training set, giving further assurance in the validity of our findings. Thirdly, with its 15.5B parameters it is likely considerably smaller than GPT-3.5 and GPT-4[5], allowing us to compare performance at different model scales.

**Controlled baselines:** We compare LINC to three baselines, which we call Naïve, Scratchpad, and Chain-of-Thought (CoT), as illustrated in Fig. 2. In the Naïve baseline, the model is given the natural language premises and is asked to directly generate the label (True/False/Uncertain).

In the Scratchpad baseline (Nye et al., 2021), the model is asked to first generate FOL expressions corresponding to the premises, and then generate the label. This baseline is thus an ablation of LINC, where we use the LLM instead of Prover9 as the logic solver. Finally, in the CoT baseline, we use the standard technique of CoT prompting (Wei et al., 2022; Kojima et al., 2022; Wang et al., 2023b), where the model is asked to generate step-by-step natural language reasoning to arrive at the conclusion. The prompts we use for all approaches can be found in Appendix D.

## 4 Results & Discussion

Our main results are shown in Figure 3. Each bar represents either LINC or one of the three baselines, while each group of bars indicates the language model used (in {StarCoder+, GPT-3.5, GPT-4}).

We note first that in the FOLIO domain (Figure 3a), StarCoder+—the smallest model we experiment with—benefits the most from LINC, achieving a mean accuracy that is 14.2 points higher than the closest controlled baseline (56.0% vs. 41.8% with CoT). We find that Scratchpad, where the intermediate logical formulae are still generated but the call to the symbolic solver is ablated and replaced by the model's own prediction, does not appear to benefit performance; for StarCoder+, neither Scratchpad nor the Naïve baseline perform better than simply deterministically predicting the most common label ("Uncertain"). For GPT-3.5, the trend is similar, although the gap between LINC and the closest baseline shrinks (62.6% vs. 54.9%

---

[2]We use gpt-3.5-turbo-16k-0613 and gpt-4-0613.

[3]https://huggingface.co/bigcode/starcoderplus

[4]See https://huggingface.co/spaces/bigcode/in-the-stack and https://huggingface.co/spaces/bigcode/search.

[5]Although the exact size of these models has not been made public, their common predecessor GPT-3 was known to have 175B parameters; see Brown et al. (2020).

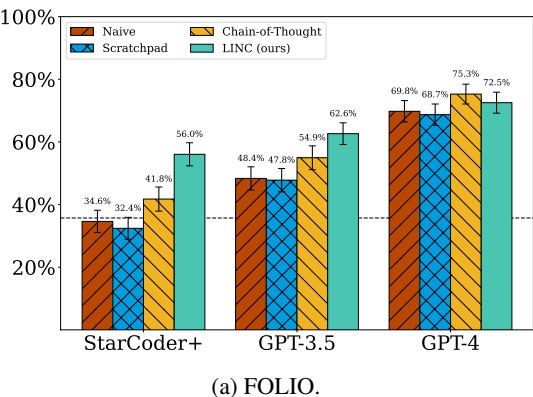

(a) FOLIO.

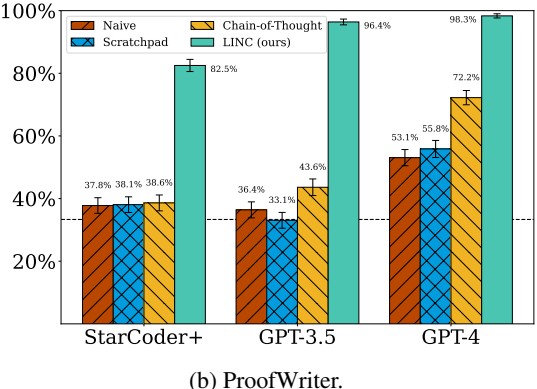

(b) ProofWriter.

Figure 3: Results of each model on the FOLIO and ProofWriter datasets. Accuracies are for bootstrapped 10-way majority vote for all models. Error bars are ±1 bootstrapped standard deviation. Dotted, black line is the accuracy obtained by always guessing the most common label in the dataset.

average accuracies). For GPT-4, the trend reverses: `LINC` underperforms `CoT`. However, we perform a McNemar's test (McNemar, 1947) to get the p-value on this GPT-4 `LINC` vs. `CoT` comparison, and we find that the difference is not significant ($p = 0.58$). Meanwhile, for our balanced subset of ProofWriter, we see significant performance gains across the board (Figure 3b); particularly so for GPT-3.5 and GPT-4, which achieve mean accuracies of 96.4% and 98.3% when paired with `LINC`.

In light of the high accuracies obtained with `LINC` on ProofWriter, we offer two plausible reasons why `LINC` is particularly favorable on this dataset. Firstly, ProofWriter is—unlike FOLIO—completely synthetically generated, with relatively short sentences, perhaps lending itself particularly well to being formalized in FOL. However, it is noteworthy that the `Scratchpad` mode does not seem to improve performance over the `Naïve` baseline, indicating that even if the NL-to-FOL task were particularly easy in this domain, this is not something that the model is itself capable of leveraging to improve its predictions. The second reason might be that the baseline strategies struggle in this *out-of-distribution* setting, in which the model must generalize to a larger set of premises (with potentially longer deductive chains) than those found in the prompt. This distribution shift makes it harder for the model to ignore irrelevant premises in the question and carry out all deductive chains correctly. Meanwhile, with `LINC`, the symbolic solver robustly handles irrelevant premises and long deductive chains, since the LLM only needs to translate each sentence into FOL.

To test this last explanation further, we plot each model's performance across ProofWriter as a func-

tion of the necessary proof depth in Figure 4. We note first that StarCoder+'s performance remains flat and close to chance with all three baseline methods (Figure 4a). Meanwhile, with `LINC` the performance remains far above chance, although it drops somewhat as necessary proof depth increases; this performance drop suggests that StarCoder+ struggles somewhat with the NL-to-FOL translation task as the problem at hand gets larger. For GPT-3.5, all baselines perform above chance at proof depth 0 (i.e., where the conclusion can immediately be reached from the premises), but then quickly drop back down (Figure 4b). While `Chain-of-Thought` prompting allows the model to complete some depth-1 tasks, even this strategy then performs equivalently to chance (within 1 standard deviation) for higher depths. When augmented with `LINC`, however, GPT-3.5 is able to achieve near-perfect performance across all proof depths, providing evidence for the scalability of this approach to longer deductive chains. Finally, for GPT-4 we observe much stronger performance from the baselines; in particular, `CoT` performs above chance for all proof depths, and all baselines perform well for shallow proofs (Figure 4c). However, even with GPT-4 the performance drops as the necessary proof depth increases with every configuration except for `LINC`, which performs near or at ceiling through the maximum proof depth available in the dataset.

## 5 Error Analysis

Having established that `LINC` can improve performance in many settings, we now move on to our final research question: *How do the failure modes of* `LINC` *compare to those of in-context reasoning*

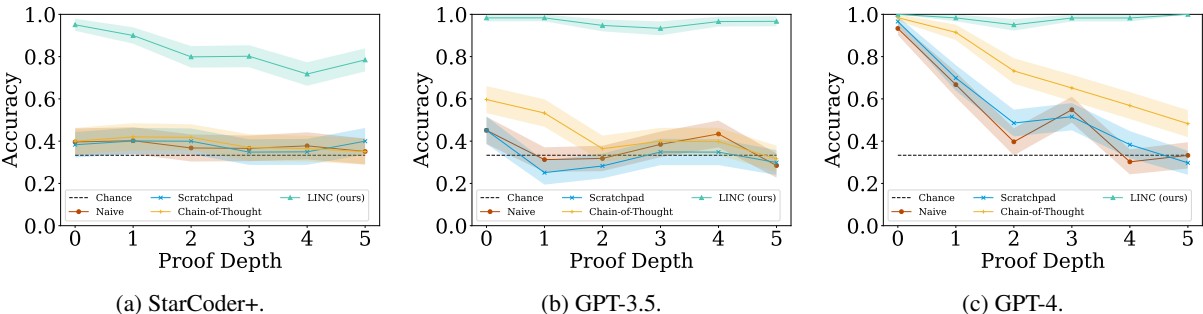

Figure 4: Accuracy per necessary proof depth in ProofWriter. Accuracies reported are for bootstrapped 10-way majority vote, and shaded areas cover ±1 bootstrapped standard deviation. Black, dotted lines reflect the expected success rate of guessing a random label, which is 1/3 in all subsets per our experiment design.

*methods*? We focus on comparing GPT-4+CoT vs. GPT-4+LINC on FOLIO, since their overall performance is very similar (75.3% vs. 72.5% average accuracy). We leave an analysis of StarCoder+'s predictions on FOLIO to Appendix G.

### 5.1 Qualitative Analysis

Qualitatively, we find that LINC and CoT have completely different failure modes. We give a high-level overview and abbreviated examples of each failure mode here, leaving full detailed examples to Appendix E.

First, we detail the failure modes for LINC:

**L1: FOL fails to capture implicit information not mentioned in the premises.** Often, there is obvious information not explicitly listed in the premises that is necessary to explicitly encode in FOL in order to successfully make a desired deduction. For example, in the snippet below one must encode in FOL the implicit assumption that Harry is a person (`Person(Harry)`).

Premise 1: When a person reads a book, that person gains knowledge.

FOL: all x. all y. (Person(x) & Reads(x, y) & Book(y) -> Gains(x, Knowledge))

Premise 2: Harry read the book "Walden" by Henry Thoreau.

FOL: Reads(Harry, Walden)

Conclusion (Prover9: Uncertain): Harry gains knowledge.

FOL: Gains(Harry, Knowledge)

**L2: FOL fails to capture information explicitly mentioned in the premises due to the choice of representation.** Even when information is explicitly written in the premises, the choice of how the NL is represented in FOL can lead to lost information. In the example below, the fact that Heinrich was a Nazi German politician is captured by

one symbol `NaziGermanPolitician`, causing the information that he was independently Nazi, German, or a politician to be lost. As a result, LINC predicted `Uncertain` instead of the ground truth label `True`.

Premise: Heinrich Scmidt was a Nazi German politician.

FOL: NaziGermanPolitician (HeinrichSchmidt)

Conclusion (Prover9: Uncertain): Heinrich Schmidt was German.

FOL: German(HeinrichSchmidt)

**L3: FOL contains syntax errors.** Across all generations, we find that the FOL expressions sometimes contain syntax errors: 38% for StarCoderPlus, 24% for GPT-3.5, and 13% for GPT-4. The most common error is that the same symbol is used with multiple arities. As an example, if `Summer(July4)` and `Loves(Alex, Summer)` were both present in a FOL translation, `Summer` would have a multiple arity violation.

Next, we detail three failure modes for CoT:

**C1: CoT concludes something different than it suggests.** One common mistake we often see with the CoT approach is that it generates a chain of reasoning indicating uncertainty, but then asserts that the conclusion is false. An example of this is shown below, where GPT-4 suggests there is "no evidence" for the conclusion but then deduces it false.

(GPT-4 Incorrect CoT): Let's think step by step. We want to evaluate if platypus are reptiles. We know from the premises that platypus are mammals and that they have no teeth. The fact that some mammals have teeth does not contradict the fact that platypus, although mammals, do not have teeth. Moreover, the statement about humans having teeth is unrelated to the classification of platypus. There's no evidence in the premises that would suggest platypus are reptiles. Therefore, the statement is false.

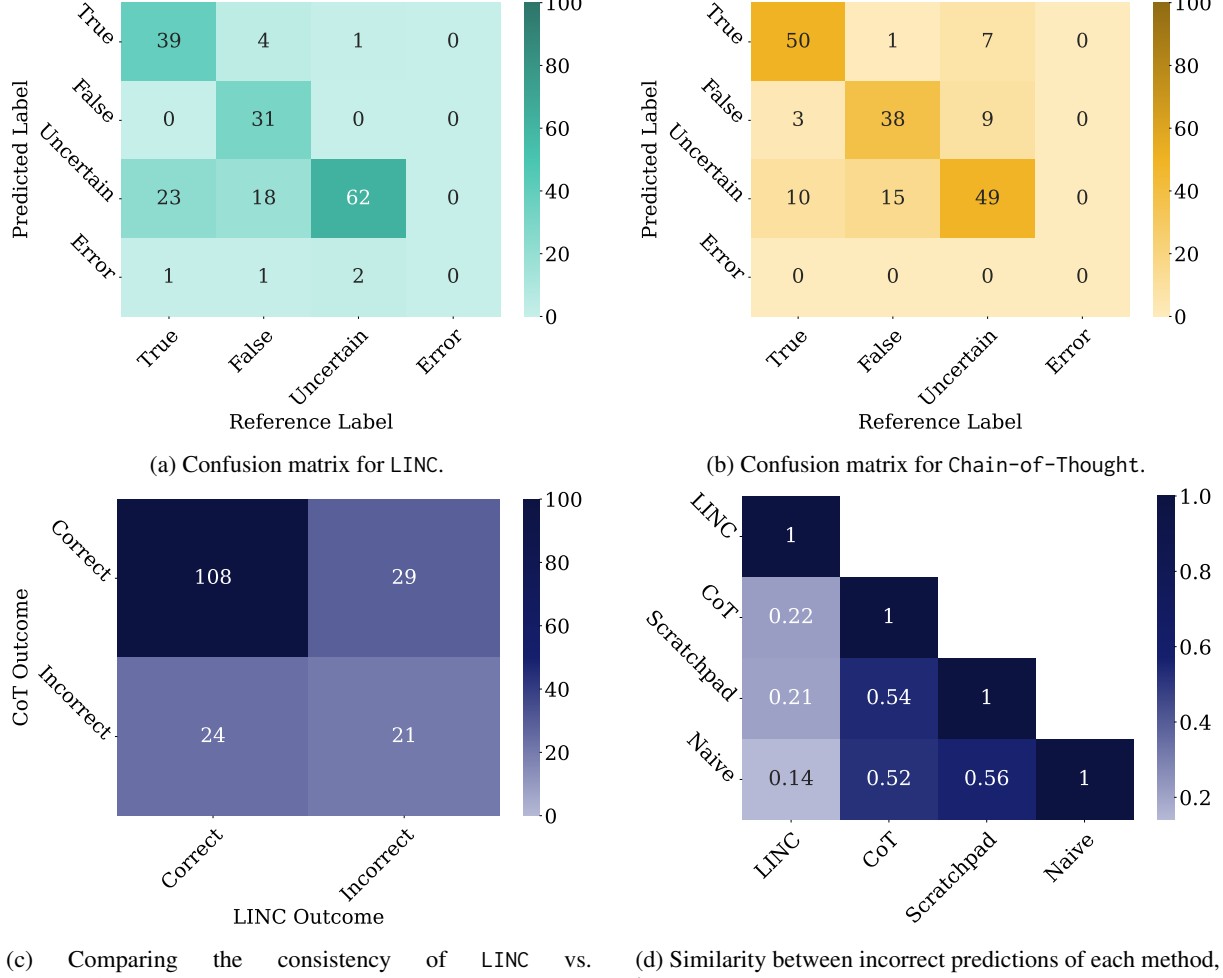

(a) Confusion matrix for LINC.

(b) Confusion matrix for Chain-of-Thought.

(c) Comparing the consistency of LINC vs. Chain-of-Thought.

(d) Similarity between incorrect predictions of each method, i.e., (A wrong == B wrong) / (A wrong or B wrong).

Figure 5: Analyzing and comparing the mistakes made by GPT-4 on the FOLIO dataset.

**C2: CoT makes incorrect logical deductions.** While there are many instances where CoT makes incorrect logical deductions, one common deduction mistake is the structure "if $B$ is true and $A \to B$, then $A$ is true". In the example shown below, we see this incorrect deduction where A is "Mike has an F1 visa" and B is "Mike needs to apply for CPT or OPT if he wants to work in the US". We note that making this type of logical mistake is often called the *fallacy of the converse*, and that humans are not immune to it, either.

Premise 1: International students in US have either an F1 visa or a J1 visa.

Premise 2: An international student in US with an F1 visa needs to apply for CPT or OPT if the student wants to work in the US.

Premise 3: Mike is an international student.

Premise 4: Mike needs to apply for CPT if he wants to work in the US.

Conclusion (Ground Truth: Uncertain): Mike has an F1 visa.

**(GPT-4 Incorrect CoT):** Let's think step by step. We want to evaluate if Mike has an F1 visa. From the premises, we know that Mike is an international student and needs to apply for CPT if he wants to work in the US. We also know that an international student in the US with an F1 visa needs to apply for CPT or OPT if the student wants to work in the US. So, it is clear that Mike has an F1 visa since he fits the criteria. Therefore, the conclusion is true.

**C3: CoT fails to find complex paths of reasoning.** We find that with all three models, CoT fails when the path of reasoning necessary to make the deduction is complex. Sometimes, CoT has difficulty getting started, and other times, it gets stuck in the middle of a reasoning chain.

### 5.2 Quantitative Analysis

**1. Compared to CoT, LINC has worse recall but better precision on True/False predictions.** To see this, we plotted the confusion matrices in Figure 5b (CoT) and Figure 5a (LINC). Looking just at the distributions of predicted labels of the two methods, we see that CoT predicts 32% True, 27%

`False`, and 41% `Uncertain`, while `LINC` predicts 24% `True`, 17% `False`, and 57% `Uncertain` (with 2% of predictions throwing an error). Notably, we observe that `LINC` predicts `Uncertain` much more frequently than `CoT` (57% vs. 41%). To understand why, note that the translation from natural language to FOL is a lossy process: recall that in L1 and L2, we saw that the information conveyed through the FOL is sometimes a subset of the information in the original premises. Removing pieces of crucial information that were on the critical path to deducing `True`/`False` may then leave an uncertain conclusion. At the same time, while the FOL translations sometimes do not retain all of the information in the NL, they rarely contain false information that was not provided in the original premises. Therefore, `LINC`'s precision when predicting `True` or `False` is very high (93%) compared to that of `CoT` (81%), but this comes at the cost of lower recall on `True`/`False` predictions (60% for `LINC` vs. 75% for `CoT`).

**2. `LINC` and `CoT` mispredict on different examples.** Earlier in Sec. 5.1, we saw that `LINC` and `CoT` exhibit different failure modes, which suggests they should fail on different examples. Indeed, we find that this is the case in our experiments on FO-LIO: Figure 5c shows a $2 \times 2$ confusion matrix which compares whether or not each method's prediction was correct. We observe that out of the $24 + 29 + 21 = 74$ samples where at least one method makes an incorrect prediction, only 21 are shared. On a closer examination of these 21 samples, we find that 16 are ambiguous or incorrect in their specification (details in Appendix E.4), so *the two methods only agree on 5 well-formed samples*. This suggests that `LINC` and `CoT` are complementary methods which fail under distinct circumstances.

**3. Mispredictions of in-context reasoning baselines are more similar to each other than they are with mispredictions of `LINC`.** As an extension of the previous analysis, we next investigate the correlation between the mispredictions of each pair of methods. To do so, we define a similarity score between two methods $A$ and $B$ as follows: Given a dataset $D$ with $N$ rows and ground truth labels $\{A_i\}_{i=1}^{N}$ and $\{B_i\}_{i=1}^{N}$ from two methods $A$ and $B$, we define

$$\text{sim}_D(A, B) \triangleq \frac{\sum_{i=1}^{N} \mathbf{1}\left[A_i = B_i \neq R_i\right]}{\sum_{i=1}^{N} \mathbf{1}\left[A_i \neq R_i \text{ or } B_i \neq R_i\right]}$$

In words, $\text{sim}_D(A, B)$ measures the number of instances where $A$ and $B$ are wrong in identical ways

vs. the number of instances where *at least one* of them is wrong.

Figure 5d shows the pairwise similarity between our four methods, highlighting that the similarity between `LINC`'s mispredictions and the other methods' mispredictions (0.14, 0.21, 0.22) is much lower than the similarity between any pair of the in-context reasoning methods (0.52, 0.54, 0.56). These results suggest that for GPT-4 on FOLIO, `LINC` is the only method we evaluate which significantly alters the ways in which the model *fails* to reason.

# 6 Related Work

**Reasoning in LLMs:** Our work contributes to the wider literature on eliciting natural language reasoning capabilities in models. Although we have focused here on comparing a neurosymbolic approach to Scratchpad (Nye et al., 2021) and `Chain-of-Thought` prompting (Wei et al., 2022; Kojima et al., 2022; Wang et al., 2023b), many other similar or related techniques have been developed in recent years; these include least-to-most prompting (Zhou et al., 2023), selection-inference (Creswell et al., 2023), backward chaining (Tafjord et al., 2022; Kazemi et al., 2023), and self-taught reasoning (Zelikman et al., 2022). Some of these techniques have been formalized under the language model cascades framework (Dohan et al., 2022).

**Semantic parsing:** The notion of a semantic parser rests on a long tradition of research (Kamath and Das, 2019) whose aim is to map fragments of natural language into useful, symbolic meaning representations (Zelle and Mooney, 1996; Zettlemoyer and Collins, 2005; Berant et al., 2013; Liang et al., 2013; Wong et al., 2023). Unlike earlier works in this tradition, we use a language model to generate the semantic parse, which is a method under active investigation in recent years (Shin and Van Durme, 2022; Drozdov et al., 2022; Lu et al., 2022; Wang et al., 2023a).

**Neurosymbolic approaches for reasoning:** Methods which combine neural networks with symbolic techniques have seen broad uptake in domains adjacent to logical reasoning, such as generating outputs consistent with a pre-existing symbolic knowledge base (Marra et al., 2019; Manhaeve et al., 2018; Zhang et al., 2023a) and performing algorithmic reasoning over symbolically grounded inputs (Ebrahimi et al., 2021; Ibarz et al., 2022;

Veličković et al., 2022). As for logical reasoning with LLMs in particular, there have been a few different proposals for when and how to best combine the LLM with a symbolic component. Zhang et al. (2022) finetune a language model to synthesize potential facts paired with likelihoods and then use a handwritten differentiable symbolic reasoner in order to deduce other facts. Weir and Van Durme (2022) relax the solver by instead training neural "entailment" models to decide if and how a given inference rule applies at each stage. Concurrently to this work, Logic-LM (Pan et al., 2023) and SATLM (Ye et al., 2023) propose neurosymbolic approaches which have much in common with LINC. However, other than the models and datasets considered, their contributions have a few key differences to ours. First, we place particular emphasis on establishing an in-depth understanding of the relative benefits and drawbacks of a neurosymbolic approach to reasoning when compared to traditional in-context reasoning strategies like Chain-of-Thought. Second, Logic-LM employs a self-refinement strategy, which has shown promise across code generation and NLP tasks (Zhang et al., 2023b; Chen et al., 2023a; Peng et al., 2023; Madaan et al., 2023; Olausson et al., 2023) but which we do not consider here. Third, SATLM studies arithmetic reasoning in addition to logical reasoning, showcasing the versatility of the neurosymbolic approach. Fourth, and finally, we use an FOL representation that we believe is easier for humans to read and models to learn. We highly encourage interested readers to study these two contemporary works in detail.

**Autoformalization:** The idea of automatically translating natural language into structured symbolic representations that programs can reason about has gained popularity in the domain of formal mathematics, leading to autoformalization systems for several theorem provers including Mizar (Wang et al., 2018, 2020), Lean 3 (Azerbayev et al., 2023), and Isabelle (Wu et al., 2022). Outside formal mathematics, autoformalization has also been applied to translating natural language into system specification languages such as temporal logic (Hahn et al., 2022; Cosler et al., 2023; Chen et al., 2023b).

**Tool usage:** Our work is heavily inspired by recent work on tool usage. The central idea in this line of research is to augment language models with external tools such as calculators, code interpreters and information retrieval systems. We further divide these works into two classes. In the first class,

the model does not need to learn how or where to invoke the tool: instead, the tool is predefined and is applied after the generation step finishes. For example, Gao et al. (2023) and Drori et al. (2022) solve mathematical reasoning tasks by generating Python programs and using the Python interpreter as the tool, Liu et al. (2023) approach physical reasoning tasks with a physical simulator as the tool, and Wong et al. (2023) tackle cognitively-inspired probabilistic reasoning tasks with Church (a probabilistic programming language) as the tool. In the second class, the model must learn to invoke the tool by itself, meaning that the model must generate explicit API calls to the tool which are then executed when those calls are decoded (Schick et al., 2023; Thoppilan et al., 2022; Yao et al., 2022; Cheng et al., 2023). Our work belongs to the former class, with the task at hand being logical reasoning and the tool available for use being a FOL solver (Prover9). We refer the reader to Mialon et al. (2023) for a more thorough survey of recent work in the tool-usage literature.

# 7   Conclusion

In this work, we present LINC: Logical Inference via Neurosymbolic Computation, a neurosymbolic approach for scalable logical reasoning with large language models. Our experiments show that LINC leads to significant performance gains in nearly every setting we consider, and that it supports generalization to settings where the model has to reason about a much larger set of premises than it is shown in the in-context learning examples. Furthermore, carrying out a quantitative and qualitative analysis of the mistakes made by LINC, we find evidence that it may complement purely in-context reasoning strategies such as Chain-of-Thought prompting, since they differ greatly in the types and frequencies of mistakes made. This work thus supports the efficacy of neurosymbolic approaches to natural language reasoning, setting the stage for continued advances in combining large language models and symbolic reasoning engines; we discuss several promising future directions in Appendix A.

# 8   Limitations

**Narrow scope of logical reasoning task considered**: In this work, we focus exclusively on one aspect of logical reasoning: predicting the truth value of a conclusion given a set of natural language premises. Here, we consider a setting where the

premises and conclusion are expressed in relatively short statements, which makes the formalization task tractable. In particular, ProofWriter's natural language statements are synthetically generated, so they can be easily and accurately parsed into FOL. FOLIO reflects a more naturalistic dataset, so we see a higher failure rate in `LINC`'s semantic parsing step. However, the formalization task becomes more difficult if the premises are in longer paragraph form, such as in question answering or contradiction detection from context passages. This is because the same piece of information can be formalized in a variety of ways, and there is a lot of information that must be pragmatically inferred to arrive at the proper conclusion.

**Generalizability of qualitative evaluation:** While we find that `LINC` and `CoT` produce complementary mistakes for our natural language reasoning task, it is unclear if this result also holds true in similar scenarios, such as the ones considered in PAL (Gao et al., 2023), Logic-LM, and SATLM. This is due to the difference in intermediate language and overall logical reasoning task. However, we hypothesize that it will and encourage future investigation in this direction.

**More sophisticated reasoning techniques**: Recent work has proposed more sophisticated techniques beyond chain-of-thought, such as tree of thoughts (Yao et al., 2023), program of thoughts, (Chen et al., 2022), or using retrieval in chain-of-thought prompting (Yasunaga et al., 2023). These have potential to improve and eliminate some of the failure modes of the traditional `CoT` method. In addition, ideas such as self-repair may also serve to improve these failure modes. It remains future work to do a more thorough investigation of the efficacy of these techniques, though there is also preliminary evidence that they still lack reasoning capabilities (Huang et al., 2023).

**Scalability**: It is unclear how well `LINC` will perform as the number of premises scales. First, one mistake in formalization can lead to an incorrect deduction, and more premises lead to a higher probability of errors. Second, in the deduction stage, while many fast algorithms (e.g., forward- and backward-chaining) exist for logical deduction, the general problem is still NP-hard. Therefore, the theorem prover may take a long time in practice.

**Other logics beyond first-order logic**: In this work, we exclusively focus on first-order logic. However, FOL is not expressive enough to handle problems requiring *higher-order* logics (Miller and Nadathur, 1986; Higginbotham, 1998). Also, in many settings it is desirable to work with *non-classical* logics (Priest, 2008; Burgess, 2009). Alternative theorem provers would be needed for such problems. A method like `LINC` can be naturally extended to those settings, but exactly how well it works there requires further investigations.

**Computational costs**: Implementing our approach with both GPT models and the StarCoder+ model requires non-trivial resources. The former requires reliance on costly API requests and the latter dedicated GPUs for inference. Especially as we use majority voting, many generations must be made for each query, increasing the computational requirements.

# 9 Acknowledgements

T.X. Olausson is supported by the Defense Advanced Research Projects Agency (DARPA) under the ASKEM program, award HR00112220042. A. Gu is supported by the National Science Foundation (NSF) Graduate Research Fellowship under Grant No. 2141064. B. Lipkin and C.E. Zhang are supported by MIT Presidential Fellowships. A. Solar-Lezama is supported by the National Science Foundation (NSF) and Intel Corporation through NSF Grant CCF:2217064. J.B. Tenenbaum is supported by AFOSR Grant #FA9550-22-1-0387 and the MIT-IBM Watson AI Lab. R.P. Levy is supported by a grant from the Simons Foundation to the Simons Center for the Social Brain at MIT.

We thank our anonymous reviewers for their insightful feedback and recommendations. We thank the members of the Computer Aided Programming, Computational Psycholinguistics, and Computational Cognitive Science groups for constructive commentary at various stages of this project. We thank Yoon Kim for helpful suggestions and comments on pieces of the initial project proposal. In addition, we thank Zhaofeng Wu and Simeng Han for discussions regarding the FOLIO dataset.

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

## A Future Directions

Of the error types we catalog in our analysis of LINC, the key opportunity for improvement is more elegant handling of naturalistic language use. While the errors observed with CoT result from faulty deductive inferences, in the case of LINC, all errors have been localized to the semantic parsing procedure. This process flows primarily unimpeded in the evaluation of the synthetic ProofWriter dataset, yet leaves room for improvement with the naturalistic FOLIO. In follow-up work, we hope to deeply explore naturalistic evaluation settings, as when data get the most messy is also where improvements become the most valuable. Here, we propose three strategies for further improvement on naturalistic settings.

First, in naturalistic communication, "obvious" information is often left out of explicit productions, left to be inferred in the "common ground" of the communicative act (Grice, 1975; Stalnaker, 2002). Implicit premise rediscovery through controlled exploration on the logical neighborhood of the existing explicit premises promises to be a powerful strategy for improving performance in underspecified settings.

Second, while a number of samples are lost to syntax errors, recent work has proposed restricting the sampling space of an LLM to that which is consistent with term expansions in a context-free-grammar (CFG) (Poesia et al., 2022). Doing so in this setting would eliminate all syntax errors.

Third, sometimes, the translation process to FOL is lossy, throwing away valuable information present in the original sentence. We propose improving the faithfulness of FOL translations by asking the LLM to translate the FOL back to natural language and comparing with the original. Forward translations that rank highly when back-translated would be those which have effectively captured the intricacies of a particular sentence's semantics.

Overall, we believe that shifting to evaluations on more naturalistic datasets, and incorporating strategies such as those presented here, will help pave the path forward for neurosymbolic approaches to formal reasoning.

## B Model Details and Parameters

We use a decoding temperature $T = 0.8$ for all models. For GPT-3.5 and GPT-4, we limit the maximum number of tokens to generate to 1024 for FOLIO and 4096 for ProofWriter (to accommo-

date for the previously mentioned larger number of premises involved in a typical question). For the StarCoder+ model, we allow generation up until the 8192 context window length, since this model is run locally. In either case, decoding is halted early whenever the stop token </EVALUATE> is produced. All local experiments were executed on a cluster equipped with NVIDIA A100 GPUs.

**GPT Models**: We use the gpt-3.5-turbo-16k-0613 and gpt-4-0613 checkpoints of the GPT-3.5 (Ouyang et al., 2022) and GPT-4 (OpenAI, 2023) models, respectively, invoking both models via the OpenAI API.

**StarCoder+**: StarCoder+ (15.5B)[6] is a version of StarCoderBase (Li et al., 2023) which has been finetuned on 600B tokens from a combination of (1) Falcon RefinedWeb (Penedo et al., 2023) (filtered version of CommonCrawl), (2) The Stack v1.2 (Kocetkov et al., 2022), and (3) a Wikipedia dataset. Its base model, StarCoderBase, is an open-access model with a GPT-2 architecture using multi-query attention (Shazeer, 2019) and fill-in-the-middle objective (Bavarian et al., 2022). StarCoderBase has a 8192 context window and is trained on $1T$ code tokens of permissively licensed text from GitHub across 80 programming languages (Li et al., 2023). We use StarCoder+ instead of StarCoderBase because it is finetuned on natural language, which should improve the performance on our task. We run StarCoder+ with bf16 precision to reduce its memory footprint.

## C FOLIO Dataset Preprocessing

We use the publicly available FOLIO dataset on https://github.com/Yale-LILY/FOLIO. We choose representative samples from the training split of the dataset to be our few-shot examples and use the validation split of the dataset in our evaluation. The testing split is not publicly available. The original dataset has 204 validation examples. However, we discovered that there are errors in 22 of the samples. We remove these samples for our evaluation and use the remaining 182 examples. The errors as follows:

- In 4 samples, one or more of the ground truth FOL expressions have unbalanced parentheses (samples 3, 109, 110, 111).

- In 8 samples, the label obtained by executing the ground-truth FOL expressions does not

---

[6] https://huggingface.co/bigcode/starcoderplus

match the provided ground truth label. We double-checked this, first by executing the FOL expressions through Prover9 and second by checking it manually. (samples 6, 28, 30, 48, 113, 115, 139, 140).

- In 10 samples, the number of premises does not match the number of FOL expressions (samples 10, 11, 12, 88, 106, 107, 108, 174, 175, 176).

The sample numbers above refer to the line index in the validation file located at https://github.com/Yale-LILY/FOLIO/blob/main/data/v0.0/folio-validation.jsonl.

## D FOLIO Few-Shot Prompts

The methodologies we investigate do not require any finetuning on domain-specific data. Instead, we use in-context learning (ICL) with pre-trained models. We prompt the model with a set of instructions and 1-8 ICL examples, which adhere to a structured text format designed to scaffold generations and ease post-processing. In particular, we begin each ICL example with each of the NL premises wrapped in an HTML-style tag <PREMISES>. . . </PREMISES> followed by the NL conclusion wrapped in <CONCLUSION>. . . </CONCLUSION>. The requisite evaluation steps for each evaluation paradigm are then outlined in a subsequent section wrapped <EVALUATE>. . . </EVALUATE>. Following the inclusion of ICL examples, a test example is added, with the <PREMISES> and <CONCLUSION> sections. Then, the <EVALUATE> tag is opened, and the LM is allowed to proceed with causal generation until the </EVALUATE> tag is generated. Upon generation of this stop token, the <EVALUATE> block is segmented for post-processing according to the method being evaluated ({naïve, scratchpad, chain-of-thought, neuro-symbolic}).

For the few-shot examples, we use samples from the publicly available FOLIO training set. We select a set of diverse samples that are balanced across labels. Since the FOLIO training set does not come with FOL expressions for the conclusions or chain of thought prompts, we manually add both for each sample. For the $k$-shot setting ($k$<8), we use the first $k$ samples from the following list of sample indices: 126, 24, 61, 276, 149, 262, 264, 684. Here, sample $i$ refers to the $i$th line in https://github.com/Yale-LILY/FOLIO/blob/

main/data/v0.0/folio-train.jsonl. We do not optimize for the choice of few-shot examples, and this is the only set of examples we evaluated with, so it is likely that there exist better choices for few-shot examples that would lead to improved performance across the board.

### D.1 FOLIO, 1-shot (baseline)

```
The following is a first-order logic (FOL)
    ↪ problem.
The problem is to determine whether the
    ↪ conclusion follows from the premises.
The premises are given in the form of a set of
    ↪ first-order logic sentences.
The conclusion is given in the form of a single
    ↪ first-order logic sentence.
The task is to evaluate the conclusion as 'True',
    ↪ 'False', or 'Uncertain' given the
    ↪ premises.

<PREMISES>
All dispensable things are environment-friendly.
All woodware is dispensable.
All paper is woodware.
No good things are bad.
All environment-friendly things are good.
A worksheet is either paper or is environment-
    ↪ friendly.
</PREMISES>
<CONCLUSION>
A worksheet is not dispensable.
</CONCLUSION>
<EVALUATE>
Uncertain
</EVALUATE>

<PREMISES>
...premises for sample here, one premise per
    ↪ line
</PREMISES>
<CONCLUSION>
...conclusion for sample here
</CONCLUSION>
<EVALUATE>
```

Listing 1: todo

### D.2 FOLIO, 1-shot (scratchpad)

```
The following is a first-order logic (FOL)
    ↪ problem.
The problem is to determine whether the
    ↪ conclusion follows from the premises.
The premises are given in the form of a set of
    ↪ first-order logic sentences.
The conclusion is given in the form of a single
    ↪ first-order logic sentence.
The task is to translate each of the premises
    ↪ and conclusions into FOL expressions, and
    ↪  then to evaluate the conclusion as 'True
    ↪ ', 'False', or 'Uncertain' given the
    ↪ premises.

<PREMISES>
```

```
All dispensable things are environment-friendly.
All woodware is dispensable.
All paper is woodware.
No good things are bad.
All environment-friendly things are good.
A worksheet is either paper or is environment-
    ↪ friendly.
</PREMISES>
<CONCLUSION>
A worksheet is not dispensable.
</CONCLUSION>
<EVALUATE>
TEXT: All dispensable things are environment-
    ↪ friendly.
FOL: all x. (Dispensable(x) ->
    ↪ EnvironmentFriendly(x))
TEXT: All woodware is dispensable.
FOL: all x. (Woodware(x) -> Dispensable(x))
TEXT: All paper is woodware.
FOL: all x. (Paper(x) -> Woodware(x))
TEXT: No good things are bad.
FOL: all x. (Good(x) -> -Bad(x))
TEXT: All environment-friendly things are good.
FOL: all x. (EnvironmentFriendly(x) -> Good(x))
TEXT: A worksheet is either paper or is
    ↪ environment-friendly.
FOL: ((Paper(Worksheet) & -EnvironmentFriendly(
    ↪ Worksheet)) | (-Paper(Worksheet) &
    ↪ EnvironmentFriendly(Worksheet)))
TEXT: A worksheet is not dispensable.
FOL: -Dispensable(Worksheet)
ANSWER: Uncertain
</EVALUATE>

<PREMISES>
...premises for sample here, one premise per
    ↪ line
</PREMISES>
<CONCLUSION>
...conclusion for sample here
</CONCLUSION>
<EVALUATE>
```

### D.3 FOLIO, 1-shot (chain of thought)

```
The following is a first-order logic (FOL)
    ↪ problem.
The problem is to determine whether the
    ↪ conclusion follows from the premises.
The premises are given in the form of a set of
    ↪ first-order logic sentences.
The conclusion is given in the form of a single
    ↪ first-order logic sentence.
The task is to translate each of the premises
    ↪ and conclusions into FOL expressions,

<PREMISES>
All dispensable things are environment-friendly.
All woodware is dispensable.
All paper is woodware.
No good things are bad.
All environment-friendly things are good.
A worksheet is either paper or is environment-
    ↪ friendly.
</PREMISES>
<CONCLUSION>
A worksheet is not dispensable.
</CONCLUSION>
```

```
<EVALUATE>
Let's think step by step. We want to evaluate if
    ↪  a worksheet is not dispensable. From
    ↪ premise 6, we know that a worksheet is
    ↪ either paper or is environment-friendly.
    ↪ If it is paper, then from premise 3, a
    ↪ worksheet is woodware, and from premise
    ↪ 2, a worksheet is dispensable. If it is
    ↪ environment-friendly, we know it is good
    ↪ from premise 5, but we know nothing about
    ↪  whether it is dispensable. Therefore, we
    ↪  don't know if a worksheet is dispensible
    ↪  or not, so the statement is uncertain.
ANSWER: Uncertain
</EVALUATE>

<PREMISES>
...premises for sample here, one premise per
    ↪ line
</PREMISES>
<CONCLUSION>
...conclusion for sample here
</CONCLUSION>
<EVALUATE>
```

### D.4 FOLIO, 1-shot (neurosymbolic)

```
The following is a first-order logic (FOL)
    ↪ problem.
The problem is to determine whether the
    ↪ conclusion follows from the premises.
The premises are given in the form of a set of
    ↪ first-order logic sentences.
The conclusion is given in the form of a single
    ↪ first-order logic sentence.
The task is to translate each of the premises
    ↪ and conclusions into FOL expressions, so
    ↪ that the expressions can be evaluated by
    ↪ a theorem solver to determine whether the
    ↪  conclusion follows from the premises.
Expressions should be adhere to the format of
    ↪ the Python NLTK package logic module.

<PREMISES>
All dispensable things are environment-friendly.
All woodware is dispensable.
All paper is woodware.
No good things are bad.
All environment-friendly things are good.
A worksheet is either paper or is environment-
    ↪ friendly.
</PREMISES>
<CONCLUSION>
A worksheet is not dispensable.
</CONCLUSION>
<EVALUATE>
TEXT: All dispensable things are environment-
    ↪ friendly.
FOL: all x. (Dispensable(x) ->
    ↪ EnvironmentFriendly(x))
TEXT: All woodware is dispensable.
FOL: all x. (Woodware(x) -> Dispensable(x))
TEXT: All paper is woodware.
FOL: all x. (Paper(x) -> Woodware(x))
TEXT: No good things are bad.
FOL: all x. (Good(x) -> -Bad(x))
TEXT: All environment-friendly things are good.
FOL: all x. (EnvironmentFriendly(x) -> Good(x))
```

```
TEXT: A worksheet is either paper or is
    ↪ environment-friendly.
FOL: ((Paper(Worksheet) & -EnvironmentFriendly(
    ↪ Worksheet)) | (-Paper(Worksheet) &
    ↪ EnvironmentFriendly(Worksheet)))
TEXT: A worksheet is not dispensable.
FOL: -Dispensable(Worksheet)
</EVALUATE>

<PREMISES>
...premises for sample here, one premise per
    ↪ line
</PREMISES>
<CONCLUSION>
...conclusion for sample here
</CONCLUSION>
<EVALUATE>
```

## E  FOLIO Error Analysis

### E.1  Ambiguity of "Either" statements

Depending on the context, the phrase "either x or y" could mean x XOR y, x OR y, or be ambiguous. Throughout our experiments, we found that models had many creatively incorrect ways of translating these statements. One reoccurring error was that statements that clearly intended x XOR y (such as, "an animal is either a rabbit or a squirrel") were translated into x OR y. We tried to account for this into account by including multiple samples with this construct in the few shot examples (see Sec. D.4). However, the models still handle this construct inconsistently and incorrectly.

In addition, we find that throughout the FOLIO dataset, by matching the natural language premises to the FOL premises, we find no consistent or predictable pattern as to how "either x or y" statements are translated. For example, "an animal is either a rabbi or a squirrel" is translated as all x. Rabbit(x) | Squirrel(x), while we believe this instance should clearly be XOR. Therefore, we believe that some of these samples are inherently ambiguous or malformed.

To highlight model behavior on these examples, four representative examples from the FOLIO validation set are shown below; examples have multiple translations because we used temperature $T = 0.8$. Here, *Correct/Incorrect* indicate whether the translations match the ground truth (which doesn't necessarily match how we would translate it).

```
Premise: an animal is either a rabbit or a
    ↪ squirrel
(Ground Truth) Translation: all x. (Rabbit(x) |
    ↪ Squirrel(x))
(Correct) Translation 1 (GPT-3.5): all x. (
    ↪ Animal(x) -> (Rabbit(x) | Squirrel(x)))
```

```
(Incorrect) Translation 2 (StarCoderPlus): ((
    ↪ Rabbit(Animal) & -Squirrel(Animal)) | (-
    ↪ Rabbit(Animal) & Squirrel(Animal)))
(Incorrect) Translation 3 (GPT-4): all x. ((
    ↪ Animal(x) & Rabbit(x)) | (Animal(x) &
    ↪ Squirrel(x)))

Premise: a person either studys or teaches
(Ground Truth) Translation: all x. (Study(x) |
    ↪ Teaches(x))
(Incorrect) Translation 1 (StarCoderPlus):
    ↪ Studys(Person) | Teaches(Person)
(Incorrect) Translation 2 (StarCoderPlus): ((
    ↪ Study(Person) & -Teach(Person)) | (-Study
    ↪ (Person) & Teach(Person)))
(Correct) Translation 3 (GPT-4): all x. (Studies
    ↪ (x) | Teaches(x))

Premise: A man is either kind or evil.
(Ground Truth) Translation: all x. (Kind(x) & -
    ↪ Evil(x)) | (-Kind(x) & Evil(x))
(Incorrect) Translation 1 (GPT-3.5): ((Man(x) & -
    ↪ Kind(x)) | (-Man(x) & Evil(x)))
(Incorrect) Translation 2 (StarCoderPlus): Kind(
    ↪ AMan) | Evil(AMan)
(Incorrect) Translation 3 (StarCoderPlus): (Kind
    ↪ (x) | Evil(x))

Premise: Ben is either from The Simpsons or
    ↪ funny.
(Ground Truth) Translation: (Simpsons(Ben) & -
    ↪ Funny(Ben)) | (-Simpsons(Ben) & Funny(Ben
    ↪ ))
(Correct) Translation 1 (StarCoderPlus): ((
    ↪ Simpsons(Ben) & -Funny(Ben)) | (-Simpsons
    ↪ (Ben) & Funny(Ben)))
(Incorrect) Translation 2 (GPT-3.5): (
    ↪ FromTheSimpsons(Ben) | Funny(Ben))
(Incorrect) Translation 3 (GPT-4):
    ↪ FromTheSimpsons(Ben) | Funny(Ben)
```

### E.2  GPT-4 LINC Failure Modes

**L1: FOL fails to capture implicit information not mentioned in the premises.** Three examples of errors from the FOLIO validation set are shown below. The first two occurring in both GPT-3.5 and GPT-4, and the latter only occurs in GPT-3.5 and interestingly, is correct in GPT-4. In Example 1, to make the correct conclusion, we must encode in FOL that Harry is a person (Person(Harry)) and that Walden is a book (Book("Walden")). Harry being a person is implicit, but "Walden" being a book is explicitly mentioned in premise 4 but fails to be explicitly encoded by the model. In Example 2, we must encode that KiKi is an animal to make the correct deduction. One can argue that this example is ambiguous, but from the context, most would make this inference. In Example 3, we need a clause that says LGA and LGA are the same airport (SameAirport(LGA, LGA)).

```
Example 1 (GPT-4)
```

```
Premises:
Books contain tons of knowledge.
When a person reads a book, that person gains
    ↪ knowledge.
If a person gains knowledge, they become smarter.
    ↪
Harry read the book "Walden" by Henry Thoreau.

Conclusion (Ground Truth: True)
Harry is smarter than before.

FOLs:
all x. (Book(x) -> Contains(x, Knowledge))
all x. all y. (Person(x) & Reads(x, y) & Book(y)
    ↪ -> Gains(x, Knowledge))
all x. (Gains(x, Knowledge) -> Smarter(x))
Reads(Harry, Walden)

Conclusion (Prover9 Deduction: Uncertain)
Smarter(Harry)

Example 2 (GPT-4)
Premises:
If an animal barks, then Jane does not like it.
All dogs are animals that bark.
If an animal jumps, then Jane likes it.
If an animal has legs, then it jumps.
If an animal is terricolous, then it has legs.
If KiKi neither jumps nor has legs, then KiKi is
    ↪ terricolous or has legs.

Conclusion (Ground Truth: True)
KiKi neither barks nor is a dog.

FOLs:
all x. (Animal(x) & Barks(x) -> NotLike(Jane, x)
    ↪ )
all x. (Dog(x) -> Animal(x) & Barks(x))
all x. (Animal(x) & Jumps(x) -> Like(Jane, x))
all x. (Animal(x) & HasLegs(x) -> Jumps(x))
all x. (Animal(x) & Terricolous(x) -> HasLegs(x)
    ↪ )
(-Jumps(KiKi) & -HasLegs(KiKi)) -> (Terricolous(
    ↪ KiKi) | HasLegs(KiKi))

Conclusion (Prover9 Deduction: Uncertain)
-Barks(KiKi) & -Dog(KiKi)

Example 3 (GPT-3.5):
Premises:
Susan flies to LGA airport.
The departure and arrival can not be the same
    ↪ airport.
John flies from LGA airport.

Conclusion (Ground Truth: False)
Susan flies from LGA airport.

FOLs:
Flies(Susan, LGA)
all x. all y. (Departure(x) & Arrival(y) & -
    ↪ SameAirport(x, y))
Flies(John, LGA)

Conclusion (Prover9 Deduction: Uncertain):
Flies(Susan, LGA)
```

**L2: FOL fails to capture information explicitly mentioned in the premises due to the** **choice of representation.** We show four illustrative examples: the first two examples occur in both GPT-3.5 and GPT-4, while the last two examples only occur in GPT-3.5 and are correct when using GPT-4. The first example is described in the main text. In example 2, we know that Barutin Cove is a cove, but this lost information is not encoded in the FOL, so we cannot conclude Cove(BarutinCove), although this is crucial information for making the deduction. In Example 3, we must deduce that Tom's license plate is a license plate (FromIstanbul(TomPlate)), which is obvious but is not encoded by the model. In the final example, the idea that an object has four sides is expressed both as FourSided(x) and FourSides(x), so the model is unable to make the correct deduction.

```
Example 1 (GPT-4)
Premises:
Heinrich Scmidt was a Nazi German politician.
Heeinrich Scmidt was also a member of the
    ↪ Prussian State Parliament and the Nazi
    ↪ Reichstag.

Conclusion (Ground Truth: True):
Heinrich Schmidt was either German or Russian.

FOLs:
GermanPolitician(HeinrichSchmidt) & Nazi(
    ↪ HeinrichSchmidt)
MemberPrussianStateParliament(HeinrichSchmidt) &
    ↪ MemberNaziReichstag(HeinrichSchmidt)

Conclusion (Prover9 Deduction: Uncertain):
German(HeinrichSchmidt) | Russian(
    ↪ HeinrichSchmidt)

Example 2 (GPT-4)
Premises:
Barutin Cove is named after the Bulgarian
    ↪ settlement of Barutin.
Barutin Cove is on the southwest coast of Snow
    ↪ Island.
The South Shetland Islands include Snow Island,
    ↪ Greenwich Island, and Deception Island.
The South Shetland Islands are considered part
    ↪ of Antarctica.
If A is located in B and B is located in C, then
    ↪ A is located in C.

Conclusion (Ground Truth: True)
There is at least one cove in Antarctica named
    ↪ after a place in Bulgaria.

FOLs:
NamedAfter(BarutinCove, Barutin) & In(Barutin,
    ↪ Bulgaria)
On(BarutinCove, SnowIsland)
In(SnowIsland, SouthShetlandIslands) & In(
    ↪ GreenwichIsland, SouthShetlandIslands) &
    ↪ In(DeceptionIsland, SouthShetlandIslands)
In(SouthShetlandIslands, Antarctica)
all x. all y. all z. ((In(x, y) & In(y, z)) ->
```

```
      ↪ In(x, z))

Conclusion (Prover9 Deduction: Uncertain)
exists x. (Cove(x) & In(x, Antarctica) & exists
      ↪ y. (In(y, Bulgaria) & NamedAfter(x, y)))

Example 3 (GPT-3.5)
Premises:
All vehicle registration plates in Istanbul
      ↪ begin with the number 34.
Plates that do not begin with the number 34 are
      ↪ not from Istanbul.
Joe's vehicle registration plate is from
      ↪ Istanbul.
Tom's license plate begins with the number 35.
If a license plate begins with the number 35,
      ↪ then it does not begin with the number
      ↪ 34.

Conclusion (Ground Truth: False):
Tom's license plate is from Istanbul.

FOLs:
all x. (VehicleRegistrationPlate(x) ->
      ↪ BeginsWith(x, 34))
all x. (-BeginsWith(x, 34) -> -FromIstanbul(x))
FromIstanbul(JoePlate)
BeginsWith(TomPlate, 35)
all x. (LicensePlate(x) & BeginsWith(x, 35) -> -
      ↪ BeginsWith(x, 34))

Conclusion (Prover9 Deduction: Uncertain):
FromIstanbul(TomPlate)

Example 4 (GPT-3.5)
Premises:
All squares have four sides.
All four-sided things are shapes.

Conclusion (Ground Truth: True):
All squares are shapes.

FOLs:
all x. (Square(x) -> FourSides(x))
all x. (FourSided(x) -> Shape(x))

Conclusion (Prover9 Deduction: Uncertain)
all x. (Square(x) -> Shape(x))
```

**L3: FOL contains syntax errors.** This is one reason that we observe majority voting helps significantly for neurosymbolic methods, as multiple generations increase the likelihood that at one least one FOL representation successfully compiles. As stated in the main text, the most common error is that the same symbol is used with multiple arities. Two examples of this are shown below: in the first example, the primitive Summer is both used as a function representing that a year has a summer (in the first premise) and as a constant representing that a student loves summer. A similar phenomenon occurs in the second example with the symbol `Badults`.

```
Example 1
```

```
Premises:
There are four seasons in a year: Spring, Summer,
      ↪ Fall, and Winter.
All students who want to have a long vacation
      ↪ love summer the most.

FOLs:
all x. (Year(x) -> (Spring(x) | Summer(x) | Fall
      ↪ (x) | Winter(x)))
all x. (Student(x) & WantsLongVacation(x) ->
      ↪ Loves(x, Summer))

Prover9 Error:
The following symbols are used with multiple
      ↪ arities: Summer/1, Summer/0.

Example 2
Premises:
Badults is a British Sitcom series, starring
      ↪ members of Pappy's.
Badults piloted in July 2013 on BBC Three.

FOLs:
exists x. (Badults(x) & BritishSitcom(x) &
      ↪ StarringMembersOf(x, Pappys))
Piloted(Badults, July2013, BBCThree)

Prover9 Error:
The following symbols are used with multiple
      ↪ arities: Badults/1, Badults/0.
```

While rarer, there are also syntax errors. For illustration, two examples from GPT-4 are shown below:

```
Premise: Ailton Silva foes not play for a
      ↪ football club.
FOL: -Exists x. (FootballClub(x) & PlaysFor(
      ↪ AiltonSilva, x))
Error: Exists should not be capitalized

Premise: The security deposit can be either
      ↪ equal to one month's rent or more.
FOL: all x. all y. (Rent(x, y) -> (Deposit(x) =
      ↪ y | Deposit(x) > y))
Error: > is not allowed in Prover9's FOL
```

**L4: There are mistakes in the FOL translation.** For GPT-4, there are very few translation mistakes, but we still find a couple instances where GPT-3.5 fails to capture the semantic meaning of the NL examples properly. Two examples are shown here:

```
Premise: Platypus have no teeth
Incorrect FOL: -NoTeeth(Platypus)

Premise: Imperium doesn't have a feud with a
      ↪ professional wrestling stable that
      ↪ includes Ivy Nile.
Incorrect FOL: -all x. (Includes(x, IvyNile) -> -
      ↪ HasFeud(Imperium, x))
```

### E.3  GPT-4 CoT Failure Modes

**C1: CoT concludes something different than it suggests.** We show two examples of this below

(both of which LINC solves correctly).

```
Example 1
Premises:
Plungers suck.
Vacuums suck.
Vampires suck.
Space is a vacuum.
A duster is a household appliance that doesn't
    ↪ suck.

Conclusion:
Space is a vampire.

(GPT-4 Incorrect CoT): Let's think step by step.
    ↪  We want to evaluate if Space is a
    ↪ vampire. We know that Space is a vacuum
    ↪ and that vacuums suck. However, while it'
    ↪ s true that vampires also suck, this
    ↪ doesn't mean that everything that sucks
    ↪ is a vampire. Therefore, there is no
    ↪ reason to believe that Space is a vampire
    ↪  based on these premises, so the
    ↪ statement is false.
ANSWER: False

Example 2
Premises:
Some mammals have teeth.
Platypus have no teeth.
Platypus are mammals.
Humans have teeth.

Conclusion:
Platypus are reptiles.

(GPT-4 Incorrect CoT): Let's think step by step.
    ↪  We want to evaluate if platypus are
    ↪ reptiles. We know from the premises that
    ↪ platypus are mammals and that they have
    ↪ no teeth. The fact that some mammals have
    ↪  teeth does not contradict the fact that
    ↪ platypus, although mammals, do not have
    ↪ teeth. Moreover, the statement about
    ↪ humans having teeth is unrelated to the
    ↪ classification of platypus. There's no
    ↪ evidence in the premises that would
    ↪ suggest platypus are reptiles. Therefore,
    ↪  the statement is false.
```

**C2: CoT makes incorrect logical deductions.**

```
Example 1
Premises:
International students in US have either an F1
    ↪ visa or a J1 visa.
An international student in US with an F1 visa
    ↪ needs to apply for CPT or OPT if the
    ↪ student wants to work in the US.
Mike is an international student.
Mike needs to apply for CPT if he wants to work
    ↪ in the US.

Conclusion (Ground Truth: Uncertain):
Mike has an F1 visa.

(GPT-4 Incorrect CoT): Let's think step by step.
    ↪  We want to evaluate if Mike has an F1
    ↪ visa. From the premises, we know that
    ↪ Mike is an international student and
```

```
    ↪ needs to apply for CPT if he wants to
    ↪ work in the US. We also know that an
    ↪ international student in the US with an
    ↪ F1 visa needs to apply for CPT or OPT if
    ↪ the student wants to work in the US. So,
    ↪ it is clear that Mike has an F1 visa
    ↪ since he fits the criteria. Therefore,
    ↪ the conclusion is true.

Example 2
Premises:
If people perform in school talent shows often,
    ↪ then they attend and are very engaged
    ↪ with school events.
People either perform in school talent shows
    ↪ often or are inactive and disinterested
    ↪ members of their community.
If people chaperone high school dances, then
    ↪ they are not students who attend the
    ↪ school.
All people who are inactive and disinterested
    ↪ members of their community chaperone high
    ↪  school dances.
All young children and teenagers who wish to
    ↪ further their academic careers and
    ↪ educational opportunities are students
    ↪ who attend the school.
Bonnie either both attends and is very engaged
    ↪ with school events and is a student who
    ↪ attends the school, or she neither
    ↪ attends and is very engaged with school
    ↪ events nor is a student who attends the
    ↪ school.

Conclusion (Ground Truth: Uncertain):
Bonnie performs in school talent shows often.

(GPT-4 Incorrect CoT): Let's think step by step.
    ↪  We want to evaluate if Bonnie performs
    ↪ in school talent shows often. Based on
    ↪ the last premise, if Bonnie is a student
    ↪ who attends the school and is very
    ↪ engaged with school events, this is only
    ↪ possible if she performs in school talent
    ↪  shows often, as per the first premise.
    ↪ If she doesn't attend and is not engaged
    ↪ with school events, she is an inactive
    ↪ and disinterested member of her community
    ↪ . But the fourth premise states that
    ↪ these individuals chaperone high school
    ↪ dances, meaning they cannot be students,
    ↪ which would contradict the last premise.
    ↪ Therefore, Bonnie performs in school
    ↪ talent shows often, so the statement is
    ↪ true.
```

**C3: CoT fails to find complex paths of reasoning.** We highlight two examples below: in the first example, the ground truth is false. To make this deduction, one must reason that if a Greyhound is a Boeing 707, then it is a plane, which means it is empty, which means it cannot transport multiple passengers, which means it is not an airline, which means there are no Greyhound planes, which is a contradiction. Looking at the CoT generations, the first CoT attempt gives up after failing to find any

link between Greyhound and Boeing 707. The second generation attempts to make deductions from the premises. In this case, none of the 10 CoT reasoning chains begin with the correct step of starting from the negation of the conclusion and deducing it false.

In the second example, to make the correct deduction, we need to start from the fact that Rose is young or a student. If Rose is young, then they do not teach, which means they study, which means they are a student, which means they are a human. If Rose is a student, then they are humans. Neither of the CoT generation is able to make progress on the deduction from the information that Rose is young. In addition, the first *CoT* generation also has a logical error at the last step, where it asserts that "A or False" is False when the truthness of A is uncertain.

```
Example 1
Premises:
If something can transport multiple passengers,
    ↪ then they are not empty.
All airlines can transport multiple passengers.
All planes are empty.
All Boeing 707s are planes.
If Greyhound is not an airline, then there are
    ↪ no Greyhound planes.

Conclusion (Ground Truth: False)
A Greyhound is a Boeing 707.

(GPT-4 Incorrect CoT, 1): Let's think step by
    ↪ step. We want to evaluate if a Greyhound
    ↪ is a Boeing 707. Looking at our premises,
    ↪  we have no direct or indirect
    ↪ information linking Greyhounds to Boeing
    ↪ 707s. Given the information available, we
    ↪  cannot make any certain conclusion about
    ↪  a Greyhound being a Boeing 707.
    ↪ Therefore, it is uncertain whether a
    ↪ Greyhound is a Boeing 707.

(GPT-4 Incorrect CoT, 2): Let's think step by
    ↪ step. We want to evaluate if a Greyhound
    ↪ is a Boeing 707. The first three premises
    ↪  don't provide any information about
    ↪ Greyhound or Boeing 707s. The fourth
    ↪ premise tells us that all Boeing 707s are
    ↪  planes but doesn't tell us anything
    ↪ about Greyhound. The fifth premise tells
    ↪ us that if Greyhound is not an airline,
    ↪ then there are no Greyhound planes, but
    ↪ this still doesn't imply that a Greyhound
    ↪  is a Boeing 707. Therefore, we do not
    ↪ have enough information to conclude
    ↪ whether a Greyhound is a Boeing 707 or
    ↪ not. So, the statement is uncertain.

Example 2
Premises:
All students are humans.
If someone studies, then they are a student.
A person either studies or teaches.
```

```
No young person teaches.
Jerry neither teaches nor is a manager.
Rose is young or a student.

Conclusion (Ground Truth: True)
Rose is a human or Jerry is a manager.

(GPT-4 Incorrect CoT, 1): Let's think step by
    ↪ step. We want to evaluate if Rose is a
    ↪ human or Jerry is a manager. Let's first
    ↪ look at Rose. From premise 6, we know
    ↪ that Rose is either young or a student.
    ↪ If Rose is a student, then from premise 1
    ↪  we know that Rose is a human. If Rose is
    ↪  young, we don't know from the premises
    ↪ whether she is a human. Therefore, it is
    ↪ uncertain whether Rose is a human. Now
    ↪ let's look at Jerry. From premise 5, we
    ↪ know that Jerry is not a manager.
    ↪ Therefore, Jerry is not a manager. So, it
    ↪  is uncertain whether Rose is a human,
    ↪ but we know that Jerry is not a manager,
    ↪ so the statement is false.

(GPT-4 Incorrect CoT, 2): Let's think step by
    ↪ step. We want to evaluate if Rose is a
    ↪ human or Jerry is a manager. First, we
    ↪ know that Jerry neither teaches nor is a
    ↪ manager, which means Jerry is not a
    ↪ manager. Therefore, the only way for the
    ↪ conclusion to be true is if Rose is a
    ↪ human. We know that Rose is either young
    ↪ or a student. If she's a student, since
    ↪ all students are humans, then she is a
    ↪ human. However, if Rose is young, we don'
    ↪ t have enough information to determine
    ↪ whether she's a human. Therefore, it's
    ↪ uncertain whether Rose is a human or
    ↪ Jerry is a manager.
```

### E.4 Shared mistakes between GPT-4 CoT and LINC

As shown in Fig. 5c, there are 21 common errors between GPT-4 CoT and GPT-4 LINC. After an in-depth analysis of the examples, we see that 16 of these arise due to inherent errors in the dataset:

- 2 of these samples contain the sentence fragment "Either Zaha Hadid's design style or Kelly Wearstler's design style." as a premise. This premise is likely intended to mean all design styles are one of these two styles, but this is hard for the model to grasp from just the fragment (sample 41, 42).

- 2 of these samples contain the sentence fragment "Either female tennis players at Roland Garros 2022 or male tennis players at Roland Garros 2022." as a premise (sample 43, 45).

- 2 of these samples have an ambiguous use of "either": "Ben is either from The Simpsons or

funny," which in this case ambiguously means `XOR`. We believe this sentence is an unnatural usage of "either" (sample 142, 143).

- In 4 samples, there is a name that is implicitly an animal, but this is not clear (sample 126, 127, 128, 199). Also, in samples 126-128, there is a statement "If Rock is neither a fly nor a bird, then Rock neither flies nor breathes." that should likely say "If Rock neither flies nor is a bird, ...". With the original formulation, everything could be uncertain because nothing is known in the case that Rock is a fly (and this is a reasonable interpretation).

- In 5 samples, the ground-truth FOL representation of the natural language premise is incorrect, causing the label to be incorrect (samples 29, 79, 85, 86, 87).

- There is a sample where the conclusion is likely mis-worded and should be "Barutin Cove is not located in Antarctica." instead of "Barutin is not located in Antarctica." This changes the ground truth label. (sample 121).

The remaining 5 samples are examples where both methods fail. In all cases, `CoT` fails to find the correct reasoning chain, as the premises/reasoning path is convoluted and complex. Meanwhile, `LINC` fails as follows:

- In sample 104, `LINC` generates separate symbols for `NotLiveAtHome` and `-LiveAtHome`, which are semantically equivalent.

- In sample 149, `LINC` generates FOLs that miss implicit information; i.e. that people with names Rose and Jerry are humans.

- In samples 172 and 173, `LINC` does not generate a single valid set of FOL premises because all generated expressions have a mismatched arity issue.

- In sample 193, `LINC` fails to express the conclusion correctly: the conclusion is "If Yuri is not an American basketball player, then Yuri is a basketball player.", but the FOLs outputted are either `-American(Yuri) & -BasketballPlayer(Yuri)` or `-American(Yuri) & BasketballPlayer(Yuri)`

## F Proofwriter StarCoder+ Errors

We also analyzed, qualitatively, the errors that StarCoder+ makes on the ProofWriter dataset. Since premises from the ProofWriter dataset are relatively simple and direct, lossy translation is rarer. Most errors are due to syntax or improper representations. We show five representative errors below:

```
Premise: All quiet people are round.
StarCoder+ FOL: all x. Quiet(x) -> Round(x)
Ground Truth FOL: all x. (Quiet(x) -> Round(x))

Premise: If someone is quiet and nice then they
    ↪ are cold.
Starcoder+ FOL: all x. (Quiet(x) \& Nice(x)) ->
    ↪ Cold(x)
Ground Truth FOL: all x. ((Quiet(x) \& Nice(x))
    ↪ -> Cold(x))

Premise: If something is furry then it is quiet.
Starcoder+ FOL: Furry(x) -> Quiet(x)
Ground Truth FOL: all x. (Furry(x) -> Quiet(x))

Premise: The cow eats the dog.
Starcoder+ FOL: Cow(Eats(Dog))
Ground Truth FOL: Eats(Cow, Dog)

Premise: The cow is blue.
Starcoder+ FOL: Cow(x) \& Blue(x)
Ground Truth FOL: Blue(Cow)
```

## G StarCoderPlus FOLIO Error Analysis

For StarCoder+, we see a slightly different trend. In Fig. 6a, we see the same pattern as for GPT-4, with a majority of uncertain predictions. In Fig. 6b, however, we see that `CoT` for StarCoder+ primarily predicts true. This is likely because the model was trained on much more code than text, and may not have picked up sophisticated textual chain-of-thought reasoning capabilities. In Fig. 6c, we can see that the mispredictions between `CoT` and `LINC` differ much more for StarCoder+ than GPT-4. Finally, in Fig. 6d, we see the same trends as we saw with GPT-4 but more pronounced, as the similarity between mispredictions in `LINC` and those in the baseline methods is even lower than they were for GPT-4.

## H The effect of $K$-way majority voting on `LINC` and `CoT`

In all our experiments, we use 10-way majority voting, inspired by prior work which found that Chain-of-Thought prompting benefited therefrom (Wang et al., 2023b). However, one might wonder how robust the performance gains seen with `LINC` are to the precise value of $K$. Figure 7 thus

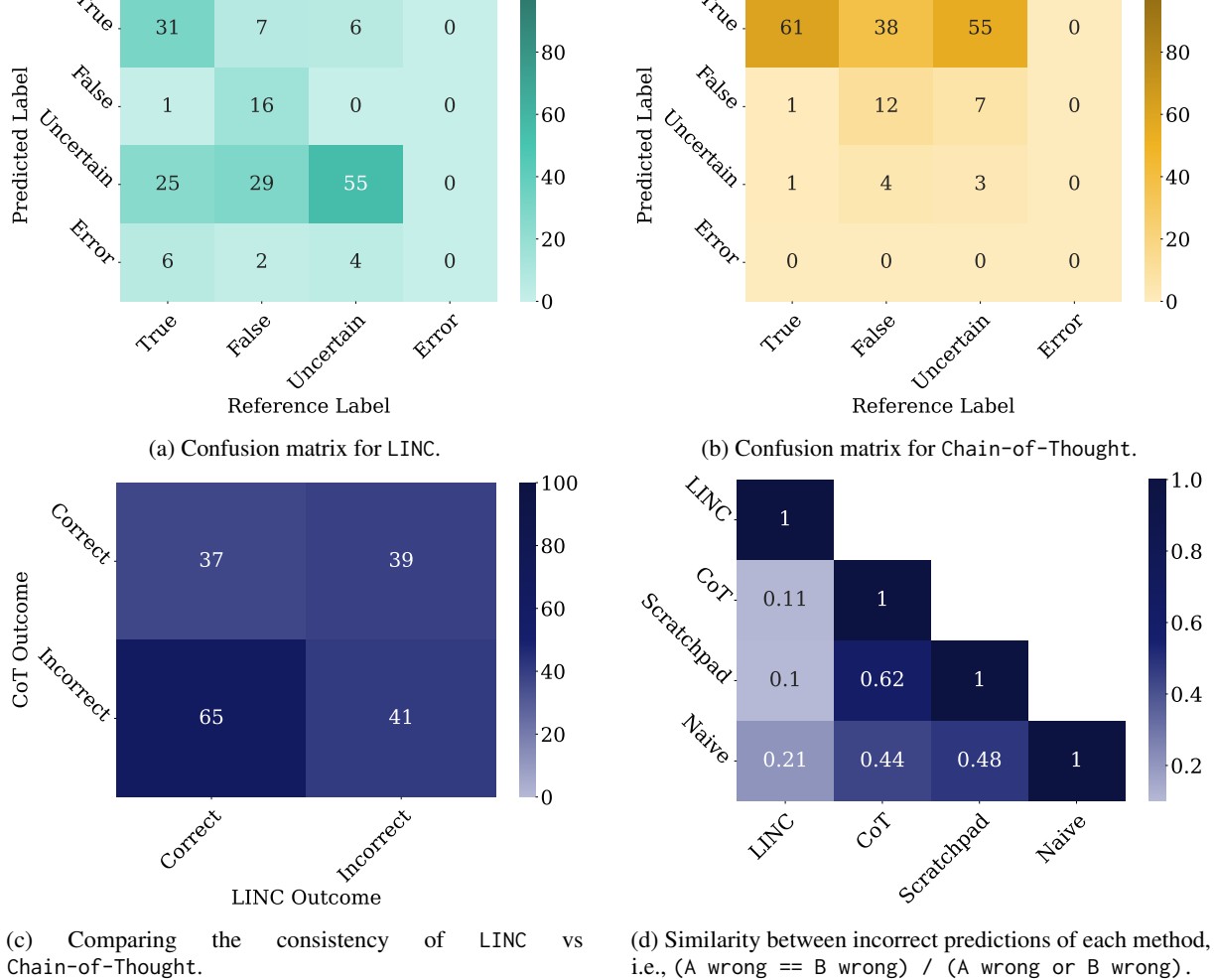

(a) Confusion matrix for LINC.

(b) Confusion matrix for Chain-of-Thought.

(c) Comparing the consistency of LINC vs Chain-of-Thought.

(d) Similarity between incorrect predictions of each method, i.e., (A wrong == B wrong) / (A wrong or B wrong).

Figure 6: Analyzing and comparing the mistakes made by StarCoder+ on the FOLIO dataset.

shows, for each model equipped with either LINC or Chain-of-Thought, how the accuracy on FO-LIO varies with $K \in \{1, 2, 3, \ldots, 10\}$. We note that, generally speaking, LINC makes good use of increased values of $K$. This is especially true for the weaker models; these are more prone to generating syntactically invalid FOL expressions, which cause the solver to return an Error. Taking the majority vote over many samples thus lessens the risk of predicting Error, which is of course always the wrong label. Notably, our results do not indicate that CoT benefits from majority voting in this domain. Future work is needed to establish how this relates to the findings in the previously mentioned prior work.

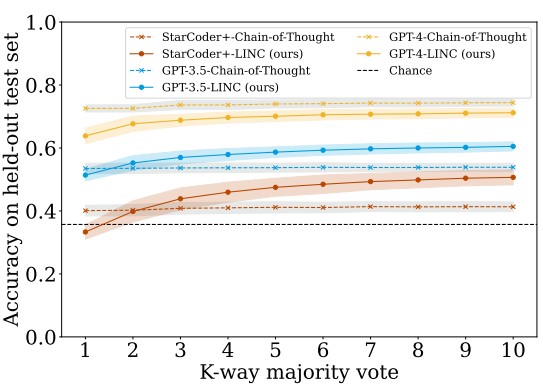

Figure 7: Accuracy on FOLIO per value of $K$ (Appendix H). Shaded areas are $\pm 1$ standard deviation over 1000 bootstrapped samples. Note that increasing $K$ benefits LINC (solid lines; shading in color) but not CoT (dashed lines; shading in gray) in our experiments on this dataset.