# OpenReview forum: "LINC: A Neurosymbolic Approach for Logical Reasoning by Combining Language Models with First-Order Logic Provers"
_EMNLP/2023/Conference — EMNLP 2023 Main_

### Official Review · Reviewer_HxCp · 2023-07-29

**Typos Grammar Style And Presentation Improvements:** The StarCoder+ model is not cited or …
**Soundness:** 4

**Excitement:**

4: Strong: This paper deepens the understanding of some phenomenon or lowers the barriers to an existing research direction.

**Missing References:**

The NL-to-FOL task is similar to the autoformalization task in mathematical reasoning, for instance:
[1] Wu Y, Jiang A Q, Li W, et al. Autoformalization with large language models[J]. Advances in Neural Information Processing Systems, 2022, 35: 32353-32368.
[2] Wang Q, Brown C, Kaliszyk C, et al. Exploration of neural machine translation in autoformalization of mathematics in Mizar[C]//Proceedings of the 9th ACM SIGPLAN International Conference on Certified Programs and Proofs. 2020: 85-98.


**Paper Topic And Main Contributions:**

This paper proposes a neurosymbolic approach of combining language models with first-order logic provers. In this framework, the LLMs first parse natural language questions into first-order logic expression, then it uses a logic prover to evaluate the expressions and obtain the final answer. The framework uses a smaller StarCoder+ and the GPT-3.5 as two backbone models and is evaluated on two datasets FOLIO and ProofWriter. The experiments show significance of the approach.

**Questions For The Authors:**

Question A: What is the pass rate of the parsed FOL expressions in the prover?
Question B: How to deal with the error propagation from parsing the FOL expressions to getting the final answer?


**Reasons To Accept:**

The performance of the proposed LINC is significant as the smaller model StarCoder+ outperforms GPT-3.5 with LINC. The analysis is comprehensive. I especially appreciate the error analysis in Section 5. The proposed method and the insightful findings are inspiring to the community. The paper is also well-written and easy to follow.

**Reasons To Reject:**

The NL-to-FOL task might not be original considering the autoformalization task in mathematical reasoning of converting natural language questions into verifiable formal proofs. Please see “Missing References” for more details.

**Reproducibility:**

3: Could reproduce the results with some difficulty. The settings of parameters are underspecified or subjectively determined; the training/evaluation data are not widely available.

**Reviewer Confidence:**

4: Quite sure. I tried to check the important points carefully. It's unlikely, though conceivable, that I missed something that should affect my ratings.

---

> ### Author Rebuttal · Authors · 2023-08-29
>
> Thank you to the reviewer for the thoughtful comments and helpful questions. We are pleased that you find our analysis comprehensive and our error analysis useful. We too hope that the method and findings could be valuable to the community. We appreciate that you point out the connections between our method and autoformalization research, and we address your questions below.
>
> **Q1: Comparison to autoformalization in mathematical reasoning**
>
> This is a great suggestion! We will certainly add a comparison to the autoformalization literature in our related work. There have been many prior works on translating NL to representations such as formal mathematical languages [1, 2, 3, 4], FOL [5, 6], temporal logic [5, 7, 8], regex [5], etc. The novelty in our LINC approach is in leveraging these autoformalization abilities to perform logical reasoning about commonsense scenarios rather than mathematical domains, and a key contribution in our paper is comparing and contrasting how LINC and CoT approach reasoning tasks. For example, one benefit of LINC is that it eliminates a class of reasoning errors introduced by CoT because it uses a theorem prover to do the logical reasoning step.
>
> [1] Wang, Q., Kaliszyk, C., & Urban, J. (2018). First experiments with neural translation of informal to formal mathematics. In Intelligent Computer Mathematics: 11th International Conference, CICM 2018, Hagenberg, Austria, August 13-17, 2018, Proceedings 11 (pp. 255-270). Springer International Publishing.
>
> [2] Wang, Q., Brown, C., Kaliszyk, C., & Urban, J. (2020). Exploration of neural machine translation in autoformalization of mathematics in Mizar. In Proceedings of the 9th ACM SIGPLAN International Conference on Certified Programs and Proofs (pp. 85-98).
>
> [3] Wu, Y., Jiang, A. Q., Li, W., Rabe, M., Staats, C., Jamnik, M., & Szegedy, C. (2022). Autoformalization with large language models. Advances in Neural Information Processing Systems, 35, 32353-32368.
>
> [4] Azerbayev, Z., Piotrowski, B., Schoelkopf, H., Ayers, E. W., Radev, D., & Avigad, J. (2023). Proofnet: Autoformalizing and formally proving undergraduate-level mathematics. arXiv preprint arXiv:2302.12433.
>
> [5] Hahn, C., Schmitt, F., Tillman, J. J., Metzger, N., Siber, J., & Finkbeiner, B. (2022). Formal specifications from natural language. arXiv preprint arXiv:2206.01962.
>
> [6] Lu, X., Liu, J., Gu, Z., Tong, H., Xie, C., Huang, J., ... & Wang, W. (2022). Parsing natural language into propositional and first-order logic with dual reinforcement learning. In Proceedings of the 29th International Conference on Computational Linguistics (pp. 5419-5431).
>
> [7] Cosler, M., Hahn, C., Mendoza, D., Schmitt, F., & Trippel, C. (2023). nl2spec: Interactively Translating Unstructured Natural Language to Temporal Logics with Large Language Models. arXiv preprint arXiv:2303.04864.
>
> [8] Chen, Y., Gandhi, R., Zhang, Y., & Fan, C. (2023). NL2TL: Transforming Natural Languages to Temporal Logics using Large Language Models. arXiv preprint arXiv:2305.07766.
>
> **Q2 (Reviewer question A): What is the pass rate of the parsed FOL expressions in the prover?**
>
> While strictly speaking Prover9 does not come with guarantees for soundness and completeness, we found in our analysis that the theorem-proving step had 100% accuracy (from FOL expressions to final answer).
>
> If we interpret the pass rate as the proportion that does not contain syntax errors, then on FOLIO it is 62% for StarCoder+, 76% for GPT-3.5, and 87% for GPT-4, as reported in Sec. 5.1.
>
> **Q3 (Reviewer question B): How to deal with the error propagation from parsing the FOL expressions to getting the final answer?**
>
> Since all errors in our LINC experiments happen in the parsing/translation stage, there is no compounding of errors in the pipeline. If there is a syntax error in a parsed FOL expression such that it fails to compile, an “Error” label will be produced as the final answer. We encourage the reviewer to check out Appendix E.1 and E.2 in addition to Sec. 5.1 for a thorough analysis of the parsing mistakes, and we are actively looking for ways to mitigate these mistakes as part of future work.
>
> **Q4: Reproducibility concerns**
>
> We strive to make our results as reproducible as possible, and we will open source all of our data (including filtered FOLIO and ProofWriter datasets) and implementations, including clearly marked scripts to reproduce each figure in the paper. Please let us know if there are any other clarifications we can include that you believe would help with reproducibility!
>
> **Q5: StarCoder+ citation**
>
> Thanks for catching this! We will cite the model in the main text.

---

### Official Review · Reviewer_iSNf · 2023-07-31

**Soundness:** 4

**Excitement:**

3: Ambivalent: It has merits (e.g., it reports state-of-the-art results, the idea is nice), but there are key weaknesses (e.g., it describes incremental work), and it can significantly benefit from another round of revision. However, I won't object to accepting it if my co-reviewers champion it.

**Missing References:**

See question A

**Paper Topic And Main Contributions:**

This paper proposes novel neuro-symbolic approach for first-order logical reasoning on natural language tasks. They deploy a large language model for parsing the natural language inputs input first-order premises, and then let FOL solvers do the logical deduction. The authors also provide extensive qualitative and quantitative analysis for the error modes and model behaviors on both proposed method and baselines.

**Questions For The Authors:**

A. Can you add discussions about non-LLM neuro-symbolic approaches in related work? I am not an expert in the field and list a few as examples. They adopt different settings as yours, but can also provide valuable perspectives for positioning your contributions. If you feel that this field is totally non-comparable, can you explain why?

Ebrahimi, Monireh, et al. "Towards bridging the neuro-symbolic gap: Deep deductive reasoners." Applied Intelligence 51 (2021): 6326-6348.
Rocktäschel, Tim, and Sebastian Riedel. "Learning knowledge base inference with neural theorem provers." Proceedings of the 5th workshop on automated knowledge base construction. 2016.

B. In a place you feel appropriate, can you list the approximate cost for running these inferences? 360 still seems a tiny number compared with the original dataset size.

**Reasons To Accept:**

The propose method is one of the first papers to combine large language models with logical solvers, so that the strong semantic understanding ability of LLMs and strict logical reasoning of FOL solvers can complement each other.

The empirical results confirm their conclusions that the proposed approach do reflect a benefit in reasoning scores.

I also appreciate the qualitative analysis on the difference in error modes of LINC compared with pure LLM reasoning (Figure 3 and section 5.1). They are not surprising but provides valuable evidence on how two paradigms work differently.

**Reasons To Reject:**

A minor reason to reject is that, although in the specific problem of FOL this approach is one of the first-of-its-kind, if we place the problem under a wider umbrella of "connecting LLMs with tools", the paradigm shows resemblance with other LLM + tool using papers. For example, if we compare with Toolformer, both approaches can be categorized in the following process:

- Prompting LLMs to convert the problem in a tool/API compatible format
- (Optional) using examples or let LLMs to correct answers
- Applying the tool/API on the converted format
- post-processing the answers.

I do acknowledge that in an era when LLMs are so powerful that people had to give up fancy model designs and reluctantly welcome LLMs into every problem, it is hard to avoid it or derive too crazy methods on overwhelmingly large LLMs. The author also made an effort to focus the paper on comparing with LLM reasoning and CoT. But I still do like to see more conceptual / theoretical comparison with other tool using papers if the "Excitement" score is to be raised.

**Reproducibility:**

4: Could mostly reproduce the results, but there may be some variation because of sample variance or minor variations in their interpretation of the protocol or method.

**Reviewer Confidence:**

4: Quite sure. I tried to check the important points carefully. It's unlikely, though conceivable, that I missed something that should affect my ratings.

---

> ### Author Rebuttal · Authors · 2023-08-29
>
> Thank you to the reviewer for your thoughtful comments and helpful questions. We are happy to see that you believe our contributions are novel and will be valuable to the community, in particular the qualitative analysis which—as you note—“provides valuable evidence on how [the] two paradigms work differently”. On the other hand, you have also contributed valuable critiques, in particular the lack of a thorough comparison between LINC and tool-augmented LLM methods more generally. We believe addressing this will help clarify to the reader where our contributions stand in comparison to contemporary work.
>
> **Q1: Conceptual/theoretical comparison with tool-using papers**
>
> We thank the reviewer for pointing out the resemblance to tool-using papers! Our work was heavily inspired by these works, and we will add a section in our related work about LLM + tool use papers. Conceptually, as you mentioned, LINC follows a very similar framework as [1-4], which solves reasoning problems into an appropriate intermediate symbolic language which is then executed to obtain an answer. Tool-based methods are quite ideal for solving reasoning tasks, as there should be no room for error in the grounded tool-use step. We compare LINC to [1-4] below:
>
> | Work | Task | Intermediate language to be executed |
> | :---: | :----: | :---: |
> | [1] (PAL) | Mathematical reasoning (GSM8K) | Python |
> | [2] | Mathematical reasoning (MATH and university benchmark) | Python (with Numpy, Sympy, and Matplotlib) |
> | [3] (Mind's Eye) | Physical reasoning (Utopia) | MuJoCo |
> | [4]  | Probabilistic reasoning (cognitively inspired tasks) | Church (probabilistic programs) |
> | LINC (ours) | Logical Reasoning (FOLIO, ProofWriter) | FOL |
>
> Another line of work [5-8], including Toolformer, proposes methods that teach themselves how and when to use tools like a calculator, an information retrieval system, a translator, etc. In these works, the model must learn when to use the tool itself, whereas in the reasoning tasks from [1-4] and LINC, it is known *a priori* which tool will be used.
>
> While the ideas from [5-8] have not yet been used in reasoning tasks, we believe there is promise in doing so. A key contribution we make is the observation that LINC and CoT make different errors (Fig. 4c), so there is high potential for future improvements by choosing whether to apply CoT or LINC on a given sample. For example, on FOLIO with GPT-4, CoT and LINC get 75% and 73% accuracies, respectively. If we could perfectly decide when to apply each technique, we would get an accuracy of 88%. We hypothesize that similar ideas of choosing when to use a direct LLM vs. when to use a tool-based approach also apply to [1-4], but they unfortunately do not contain any analysis indicating that the methods show different errors.
>
> We hope that this comparison helps with your excitement about our work! Let us know if you would like us to clarify anything else in terms of conceptual/theoretical comparison with the tool use literature.
>
> [1] Gao, L., Madaan, A., Zhou, S., Alon, U., Liu, P., Yang, Y., ... & Neubig, G. (2023). PAL: Program-aided language models. In International Conference on Machine Learning (pp. 10764-10799). PMLR.
>
> [2] Drori, I., Zhang, S., Shuttleworth, R., Tang, L., Lu, A., Ke, E., ... & Strang, G. (2022). A neural network solves, explains, and generates university math problems by program synthesis and few-shot learning at human level. Proceedings of the National Academy of Sciences, 119(32).
>
> [3] Liu, R., Wei, J., Gu, S. S., Wu, T. Y., Vosoughi, S., Cui, C., ... & Dai, A. M. (2022). Mind's Eye: Grounded Language Model Reasoning through Simulation. arXiv preprint arXiv:2210.05359.
>
> [4] Wong, L., Grand, G., Lew, A. K., Goodman, N. D., Mansinghka, V. K., Andreas, J., & Tenenbaum, J. B. (2023). From Word Models to World Models: Translating from Natural Language to the Probabilistic Language of Thought. arXiv preprint arXiv:2306.12672.
>
> [5] Schick, T., Dwivedi-Yu, J., Dessì, R., Raileanu, R., Lomeli, M., Zettlemoyer, L., ... & Scialom, T. (2023). Toolformer: Language models can teach themselves to use tools. arXiv preprint arXiv:2302.04761.
>
> [6] Thoppilan, R., De Freitas, D., Hall, J., Shazeer, N., Kulshreshtha, A., Cheng, H. T., ... & Le, Q. (2022). Lamda: Language models for dialog applications. arXiv preprint arXiv:2201.08239.
>
> [7] Yao, S., Zhao, J., Yu, D., Du, N., Shafran, I., Narasimhan, K., & Cao, Y. (2022). React: Synergizing reasoning and acting in language models. arXiv preprint arXiv:2210.03629.
>
> [8] Cheng, Z., Xie, T., Shi, P., Li, C., Nadkarni, R., Hu, Y., ... & Yu, T. (2022). Binding language models in symbolic languages. arXiv preprint arXiv:2210.02875.
>
> **Q2 (Reviewer question A): Non-LLM neuro-symbolic approaches**
>
> Thank you for bringing this oversight to our attention! The rich literature on pre-LLM neuro-symbolic work on logical reasoning was also key inspiration for this project; for the updated version, we will amend the Related Work section accordingly.
>
> Roughly speaking, pre-LLM neuro-symbolic work on logical reasoning fell into one of two buckets, both of which are quite distinct from the setting we consider:
> (1) *Neural logical reasoning*. Here one trains a neural network to perform reasoning over a chosen logic [1]. This form of neuro-symbolic reasoning integration can be seen as an instance of what is more broadly called neural algorithmic reasoning [2, 3], where the algorithm to be learned is to perform deductive reasoning. This line of work assumes that the inputs are already symbolically grounded, which is quite a different setting than in LINC and other recent work in the LLM era that has focused on reasoning over natural language inputs.
> (2) *Inference modulo logical constraints*. Here one constructs or constrains a probabilistic model, e.g., a DNN or even a simpler graphical model, so that its outputs are guaranteed to satisfy some logical constraints (e.g. matching some FOL knowledge base) [4, 5, 6]. While this allows for flexible input formats such as natural language, efficiently combining a large output space with a symbolically grounded set of constraints in a way that scales beyond toy datasets is challenging, although recent work is making some headway in this space [7]. This is a closer setting to the one we consider in our work, but the methodology is quite different: in LINC the output space of the LLM is not actually constrained, and so it does not suffer from the same scalability concerns, although of course this comes at the cost of losing all hard guarantees about the logical consistency of the outputs.
>
> In summary, the setting we study in LINC is more closely aligned to other recent work in which the LLM itself performs the symbolic grounding [8, 9, *inter alia*] than it is to most work in the older neuro-symbolic logical reasoning literature, which has traditionally focused on settings where you either have pre-existing symbolic constraints (e.g., a knowledge base) or symbolically grounded inputs (e.g., in a particular logic of your choice).
>
> [1] Ebrahimi, M., Eberhart, A., Bianchi, F., & Hitzler, P. (2021). Towards bridging the neuro-symbolic gap: Deep deductive reasoners. Applied Intelligence, 51, 6326-6348.
>
> [2] Ibarz, B., Kurin, V., Papamakarios, G., Nikiforou, K., Bennani, M., Csordás, R., ... & Veličković, P. (2022). A generalist neural algorithmic learner. In Learning on Graphs Conference (pp. 2-1). PMLR.
>
> [3] Veličković, P., Badia, A. P., Budden, D., Pascanu, R., Banino, A., Dashevskiy, M., ... & Blundell, C. (2022). The CLRS algorithmic reasoning benchmark. In International Conference on Machine Learning (pp. 22084-22102). PMLR.
>
> [4] Marra, G., Giannini, F., Diligenti, M., & Gori, M. (2019). Integrating learning and reasoning with deep logic models. In Joint European Conference on Machine Learning and Knowledge Discovery in Databases (pp. 517-532). Cham: Springer International Publishing.
>
> [5] Bach, S. H. (2015). Hinge-loss Markov random fields and probabilistic soft logic: A scalable approach to structured prediction (Doctoral dissertation, University of Maryland, College Park).
>
> [6] Manhaeve, R., Dumancic, S., Kimmig, A., Demeester, T., & De Raedt, L. (2018). Deepproblog: Neural probabilistic logic programming. Advances in neural information processing systems, 31.
>
> [7] Zhang, H., Dang, M., Peng, N., & Van den Broeck, G. (2023). Tractable control for autoregressive language generation. In International Conference on Machine Learning (pp. 40932-40945). PMLR.
>
> [8] Nye, M., Tessler, M., Tenenbaum, J., & Lake, B. M. (2021). Improving coherence and consistency in neural sequence models with dual-system, neuro-symbolic reasoning. Advances in Neural Information Processing Systems, 34, 25192-25204.
>
> [9] Gao, L., Madaan, A., Zhou, S., Alon, U., Liu, P., Yang, Y., ... & Neubig, G. (2023). Pal: Program-aided language models. In International Conference on Machine Learning (pp. 10764-10799). PMLR.
>
> **Q3 (Reviewer question B): Approximate cost for running experiments and concern about small dataset size**
>
> This is a reasonable question. The approximate cost for running these experiments was ~$600. As we note in our response to Reviewer 1NfW, we have run additional experiments using GPT-4 on ProofWriter; we repeat those results in the table below.
>
> We note that in order to carry out our extensive analysis, we required a dataset which was balanced across all combinations of depths and labels without repeating any task. This is not the case in the full ProofWriter dataset: for example, we found that there are very few depth-5 questions in ProofWriter that have the label “Uncertain”/”Unknown”. The dataset we use for our experiments has 20 questions per label (3) and proof depth (6); this results in a final dataset size of 20 * 3 * 6 = 360. We found that without cherry-picking the seed, we were unable to obtain a subset that had 30 questions per depth and label through simple rejection sampling, due to the aforementioned relative lack of “Uncertain”/”Unknown” questions. We also note that the dataset was, up to these constraints, randomly generated and would therefore likely generalize to a larger sample from the same balanced distribution.
>
> From these results, while the dataset size is small, we believe that it is sufficient to highlight our key insight on ProofWriter: while LINC achieves almost perfect accuracy across all proof depths, CoT's accuracy drops off as the proof depth increases.
>
> Together with our source code release, we will also release the code generating this balanced subset from the full dataset. Future researchers are welcome and encouraged to explore a larger number of samples within particular subsets.
>
> *GPT-4 ProofWriter results*
> | Condition | Accuracy |
> | --- | --- |
> | Baseline | 53.1% |
> | Scratchpad | 55.8% |
> | CoT | 72.2% |
> | LINC | 98.3% |

---

### Official Review · Reviewer_1NfW · 2023-08-05

**Typos Grammar Style And Presentation Improvements:** The paper is really well written.
**Soundness:** 4

**Excitement:**

4: Strong: This paper deepens the understanding of some phenomenon or lowers the barriers to an existing research direction.

**Missing References:**

Minor: Missing citation for McNemar's test.

**Paper Topic And Main Contributions:**

The authors present a framework for solving reasoning problems where an LLM converts natural language text into first-order logic. A classical theorem prover is then used to solve the reasoning problem symbolically. The authors compare their approach to chain-of-thought with GPT and StarCoder+.

**Questions For The Authors:**

 A. Why not run experiments with GPT4 on ProofWriter? (Figure 2, right side)

 B. Did you perform experiments on the paraphrased portion of ProofWriter? If so, how do the results compare?

**Reasons To Accept:**

 - They provide an in-depth exploration and comparison between their neurosymbolic approach and chain-of-thought prompting (and more broadly against methods that perform reasoning entirely in text).
 - They experiment with StarCoder+, a smaller open LLM, which facilitates reproducibility.
 - The paper is well-written and easy to understand.

**Reasons To Reject:**

 - The proposed method seems like it would perform well on tasks like ProofWriter, which was generated synthetically. But it's less clear how well it would perform on more natural language, which can be noisy. A comparison of the performance on a more realistic dataset would strenghten the paper. FOLIO is more realistic than ProofWriter, but it would be interesting to see this approach applied to a reasoning dataset without symbolic logical form labels.

**Reproducibility:**

4: Could mostly reproduce the results, but there may be some variation because of sample variance or minor variations in their interpretation of the protocol or method.

**Reviewer Confidence:**

4: Quite sure. I tried to check the important points carefully. It's unlikely, though conceivable, that I missed something that should affect my ratings.

---

> ### Author Rebuttal · Authors · 2023-08-29
>
> Thank you to the reviewer for the thoughtful comments and helpful questions. We are glad that the exploration in this work was determined to be in-depth, and that the presentation was found to be clear and easy to understand. We hope that this work can facilitate ongoing conversations and act as a reproducible marker. In addition to the use of an open source model, we will release the entirety of our code base for researchers to build upon. The link will be included in the updated manuscript.
>
> **Q1: Unclear how well our method would perform on more natural language**
>
> This is an excellent question! We agree with the reviewer that extending to more natural language is one of the core limitations of our work as it currently stands, and we believe that it is one of the most impactful directions for future work. This is one of the main motivations of our error analysis, and we believe that identifying the bottlenecks of the LINC approach will pave the way for improvements on more natural language. We provide here a discussion on this and will integrate the core information into our updated version.
>
> Indeed, while LINC’s performance on ProofWriter was quite strong, this reflects a best case scenario. ProofWriter is a synthetically generated dataset, and as such is expected to be more easily parsed back into its underlying logical structure. FOLIO, on the other hand, reflects a more naturalistic dataset, as noted by the reviewer, and as such we see a gap in performance between these two evaluations.
>
> In follow-up work, we hope to more deeply explore a more naturalistic setting and are actively looking for such benchmarks, as when things get the most messy is also where improvements become the most meaningful. Below, we propose potential solutions addressing each of the failure modes of LINC with GPT-4 on FOLIO from Appendix E.2.
>
> *L1 (lines 1365 - 1451): FOL fails to capture implicit information not present in the premises.*
>
> **Potential solution:** In naturalistic communication, “obvious” information is often left out of explicit productions, and left to be inferred by interlocutors in the “common ground”’ of the communicative act [1, 2]. We hope to directly model this phenomena as part of our approach using the following strategy. After generating the set of logic expressions explicitly presented by the problem, the model next explores the set of additional premises that might be generated to connect the sparsely connected terms thus far, filtered to those that would impact the conclusion (akin to a goal-directed strategy). The likelihood of each of these premises can then be evaluated under the LM parser conditioned on the original problem as well as new ICL examples showcasing this type of implicit premise recovery. Those with high probability of inclusion (above some to-be-determined threshold) can then be included in the premises passed to the logic prover.
>
> *L2 (lines 1452 - 1566): FOL fails to capture information explicitly mentioned in the premises due to the choice of representation.*
>
> **Potential solution:** This is a simpler case of L1 since it can be explicitly verified whether a proposed fact is explicitly stated in the premises, and all of the individual expressions in our FOL representation are relatively simple and checkable by an LM.
>
> *L3 (lines 1567 - 1629): FOL contains syntax errors.*
>
> **Potential solution:** Recent work [3, 4] has shown promise in the ability of models like GPT-4 to self-debug and correct itself from error messages, and [5] has already seen improvements from this idea in the domain of logical reasoning. Therefore, we believe that the idea of self-repair may be able to at least partially mitigate this class of errors.
>
> *L4 (lines 1630 - 1645): There are mistakes in the FOL translation.*
>
> **Potential solution:** These mistakes are rare with models like GPT-4, but can still occur from time to time. One potential method we propose for improving the faithfulness of FOL translations is to ask the LM to translate the FOL back to natural language and compare with the original.
>
> We hope that this expanded discussion surrounding the additional complexity and challenge of naturalistic evaluation, combined with an initial proposal for how we view the next steps will help make this paper more valuable for the community.
>
> [1] Grice, H. P. (1975). Logic and conversation. In D. Davidson & G. Harman (eds.), The Logic of Grammar (pp. 64–75). Dickenson.
>
> [2] Stalnaker, R. (2002). Common ground. Linguistics and philosophy, 25, 701-721.
>
> [3] Chen, X., Lin, M., Schärli, N., & Zhou, D. (2023). Teaching large language models to self-debug. arXiv preprint arXiv:2304.05128.
>
> [4] Olausson, T. X., Inala, J. P., Wang, C., Gao, J., & Solar-Lezama, A. (2023). Demystifying GPT Self-Repair for Code Generation. arXiv preprint arXiv:2306.09896.
>
> [5] Pan, L., Albalak, A., Wang, X., & Wang, W. Y. (2023). Logic-lm: Empowering large language models with symbolic solvers for faithful logical reasoning. arXiv preprint arXiv:2305.12295.
>
> **Q2 (Reviewer question A): Why not run experiments with GPT4 on ProofWriter? (Figure 2, right side)**
>
> Thanks for the suggestion, we have run this experiment and included the results below. At a high-level, the insights are as follows:
> - GPT-4 performs better than GPT-3.5 for both CoT (72.2% vs. 43.6%) and LINC (98.3% vs. 96.4%).
> - LINC still performs better than CoT across the board, and even StarCoder+ with LINC outperforms GPT-4 with CoT. (LINC: GPT-4 98.3%, GPT-3.5 96.4%, StarCoder+ 82.5%; CoT: GPT-4 72.2%, GPT-3.5 43.6%, StarCoder+ 38.6%).
> - Just like we saw with GPT-3.5 in Figure 3b, GPT-4 CoT performance steadily drops as longer deductive chains are required (98%, 92%, 74%, 65%, 57%, 48% for depths 0, 1, 2, 3, 4, 5), while LINC remains near perfect accuracy (100%, 98%, 95%, 98%, 98%, 100%).
> - On CoT, compared to GPT-3.5, gains from GPT-4 are primarily from performance on samples with low proof depth.
>
> *GPT-4 ProofWriter results*
> | Condition | Accuracy |
> | --- | --- |
> | Baseline | 53.1% |
> | Scratchpad | 55.8% |
> | CoT | 72.2% |
> | LINC | 98.3% |
>
> So, in sum, we think that these new results are consistent with the general conclusion we stated in our paper that LINC has strong advantages over the other approaches for ProofWriter, even when using GPT-4.
>
> **Q3 (Reviewer question B): Did you perform experiments on the paraphrased portion of ProofWriter? If so, how do the results compare?**
>
> While we did not perform such experiments, this is an excellent general suggestion for our follow-up work on extending LINC to more naturalistic settings, which we have looped into the discussion above. We thank the reviewer for raising this line of inquiry. Regarding specifically why we did not include the paraphrased portion of ProofWriter, it was because we found it to be less naturalistic than FOLIO, which we adopted for our initial semi-naturalistic option. See below an example from the paraphrased portion of ProofWriter, which reflects the indeed complex but still not quite natural style:
>
> ```
> sent1":"Charlie is green, but often kind, even when he is blue and cold.","sent2":"Fred rather resembles the rainbow, as he is green, red and blue; Fred is also said to be very kind.","sent3":"Harry seems to be round.","sent4":"Young people who are nice and look round are also going to be green.
> ```
>
> **Q4: Missing citation for McNemar's test**
>
> Thank you for catching this. We will add the following reference for McNemar's test:
>
> McNemar, Q. Note on the sampling error of the difference between correlated proportions or percentages. Psychometrika 12, 153–157 (1947).

---

### Meta-Review · Area_Chair_kFmC · 2023-09-17

**Recommendation:** 5

**Metareview:**

This paper proposed a semantic parser to parse natural language to first-order logic for commonsense problems. Applying this semantic parser to smaller models resolved much better results compared to LLMs (with or without CoT).

**Pros**: The paper is very well-written and easy to follow. It also provided in-depth analyses of their methods and LLM performance. The performance improvement from smaller models is also important compared to LLMs. All the reviewers appreciate these aspects of the paper.

**Cons**: It's unclear how to apply this method to more "natural language" tasks, as mentioned by 1NfW. Moreover, adding more discussion in the related work section about the comparison to Toolformer and other work in the mathematical domain of semantic parsers is going to be beneficial for the readers (HxCp, iSNf).

Overall, it's a very well-written, solid, and sound paper. The direction of combining neural models and traditional solvers is exciting to solve certain problems in commonsense domains.

---

### Decision · Program_Chairs · 2023-10-07

**Decision:**

Accept-Main

**Comment:**

This paper proposed a semantic parser to parse natural language to first-order logic for commonsense problems. Applying this semantic parser to smaller models resolved much better results compared to LLMs (with or without CoT).

**Pros**: The paper is very well-written and easy to follow. It also provided in-depth analyses of their methods and LLM performance. The performance improvement from smaller models is also important compared to LLMs. All the reviewers appreciate these aspects of the paper.

**Cons**: It's unclear how to apply this method to more "natural language" tasks, as mentioned by 1NfW. Moreover, adding more discussion in the related work section about the comparison to Toolformer and other work in the mathematical domain of semantic parsers is going to be beneficial for the readers (HxCp, iSNf).

Overall, it's a very well-written, solid, and sound paper. The direction of combining neural models and traditional solvers is exciting to solve certain problems in commonsense domains.